# Artificial dragonfly algorithm in the Hopfield neural network for optimal Exact Boolean $k$ satisfiability representation

**Ghassan Ahmed Ali**[1], **Hamza Abubakar**[2]*, **Shehab Abdulhabib Saeed Alzaeemi**[3]*, **Abdulkarem H. M. Almawgani**[4], **Adel Sulaiman**[1], **Kim Gaik Tay**[3]*

**1** College of Computer Science and Information Systems, Najran University, Najran, Saudi Arabia, **2** Department of Mathematics, Isa Kaita College of Education, Dutsin-Ma, Katsina State, Nigeria, **3** Faculty of Electrical and Electronic Engineering, Universiti Tun Hussein Onn Malaysia, Johor, Malaysia, **4** Electrical Engineering Department, College of Engineering, Najran University, Najran, Kingdom of Saudi Arabia

* zeeham4u2c@yahoo.com (HA); shehab@uthm.edu.my (SASA); tay@uthm.edu.my (KGT)

**Data Availability Statement:** We would like to clarify that the data used in this study were generated through simulation to represent Exact Boolean k-Satisfiability. No actual data was

## Abstract

This study proposes a novel hybrid computational approach that integrates the artificial dragonfly algorithm (ADA) with the Hopfield neural network (HNN) to achieve an optimal representation of the Exact Boolean $k$Satisfiability (EB$k$SAT) logical rule. The primary objective is to investigate the effectiveness and robustness of the ADA algorithm in expediting the training phase of the HNN to attain an optimized EB$k$SAT logic representation. To assess the performance of the proposed hybrid computational model, a specific Exact Boolean $k$Satisfiability problem is constructed, and simulated data sets are generated. The evaluation metrics employed include the global minimum ratio (GmR), root mean square error (RMSE), mean absolute percentage error (MAPE), and network computational time (CT) for EB$k$SAT representation. Comparative analyses are conducted between the results obtained from the proposed model and existing models in the literature. The findings demonstrate that the proposed hybrid model, ADA-HNN-EB$k$SAT, surpasses existing models in terms of accuracy and computational time. This suggests that the ADA algorithm exhibits effective compatibility with the HNN for achieving an optimal representation of the EB$k$SAT logical rule. These outcomes carry significant implications for addressing intricate optimization problems across diverse domains, including computer science, engineering, and business.

## Introduction

Satisfiability, also known as SAT, is a fundamental problem in computer science that involves determining whether a given Boolean formula can be satisfied by a set of truth values for its variables [1]. The satisfiability problem (SAT) is a well-known NP-complete problem in computer science, which has many practical applications such as circuit design, artificial intelligence, and robotics. Boolean satisfiability (Boolean SAT) is a specific case of the SAT problem in which the variables can only take on the values of true or false.

The optimization of exact Boolean SAT is important in solving many real-world problems. Therefore, developing efficient algorithms for solving the exact Boolean SAT problem has

collected from any individuals or entities, and all variables and data points were simulated for the purpose of model development and analysis. We have provided a detailed explanation of our simulation methodology in the Methods section of the paper.

**Funding:** The authors are thankful to the Deanship of Scientific Research at Najran University, Kingdom of Saudi Arabia, for funding this work under the Distinguished Research Funding program grant code (NU/DRP/SERC/12/6). The funders did not play a role in study design, data collection and analysis, decision to publish, or preparation of the manuscript.

**Competing interests:** The authors have declared that no competing interests exist.

been a topic of interest for researchers. It is a decision problem that has applications in various fields, such as artificial intelligence, verification, cryptography, and optimization. The satisfiability problem was first introduced by Stephen Cook in 1971, who proved that the problem was NP-complete, meaning that it is computationally hard to solve in general [2]. The problem involves determining whether a given Boolean formula can be satisfied by assigning truth values to its variables. A Boolean formula is a logical expression that uses Boolean operators such as AND, OR, and NOT to combine variables and their negations.

SAT is a widely used modelling framework for representing various combinatorial problems, such as Graph Coloring Problems [3], N-queen probems [4], Travelling salesmen problem [5], Scheduling problem [6] and timetabling problem [7], planning problem [8] and many more. It finds applications in various fields, including artificial intelligence, verification, cryptography, and optimization. In artificial intelligence, SAT is utilized to solve problems related to automated reasoning, planning, and knowledge representation [9, 10]. In verification, SAT is used to check the correctness of hardware and software designs [11]. In cryptography, SAT is used to break encryption algorithms and to design new ones [12]. In optimization, SAT is used to find the optimal solution to combinatorial problems such as scheduling and routing [13, 14].

In addition to its practical applications, the satisfiability problem (SAT) has influenced several related decision and optimization problems, which are known as SAT extensions. These extensions either utilize the same algorithmic techniques as SAT or employ SAT as a core engine. Some popular SAT extensions include Maximum Satisfiability (MaxSAT), Minimum Satisfiability (MinSAT), Major Satisfiability (MSAT), Model Counting (#SAT), Exact Satisfiability, Partial MAXSAT, and Quantified-Boolean Formulas (QBF). It's important to note that this list is not exhaustive, as the applications of SAT have been growing in recent years, particularly in addressing security and transportation challenges. Despite the computational complexity of SAT, which makes it a challenging problem to solve in general, efficient algorithms and heuristics have been developed to handle many instances of the problem.

Artificial neural networks (ANNs) are approaches used in machine learning and classification problems, loosely based on the biological structure of the brain and nervous systems. ANNs are powerful classifiers, and it has been mathematically proven that they can learn any mathematical function to arbitrary precision given sufficient training time and data. They can be applied to model multi-scale, nonlinear systems, as well as non-differentiable mathematical and engineering problems. ANNs are self-learning model frameworks capable of generating improved results based on available data [15]. There are various types of artificial neural networks developed for specific purposes. The Hopfield neural network (HNN) was proposed by Hopfield and Tank in 1985 as a means to model and optimize nonlinear patterns using the network's energy structure during the training and testing processes. This model has gained popularity due to its ability to interpret complex real-life problems. Hopfield neural networks (HNNs) have made significant contributions to various areas, including combinatorial optimization recognition [16–18].

One of the breakthroughs in Satisfiability logic programming and artificial neural networks was the incorporation of variants of an artificial neural network into a single model. The integration of logic programming into the Hopfield artificial neural network, referred to as HNN-SAT, was first introduced in [19]. This approach combines an artificial neural network with various logic programming problems. The aim is to utilize the optimization capabilities of the neural network to address the logical inconsistencies within the model network. After interpreting the synaptic strengths, the system relaxes into neural states that correspond to a valid or near-valid representation. The HNN models have gained wide acceptance among researchers due to their strong content addressable memory (CAM) component [20] and their

ability to converge using the Lyapunov energy function (LEF) of the HNN [21]. However, the basic HNN relies on exhaustive search (traditional method or direct search) and heuristic approaches (metaheuristics) during the training and testing stages. It has been discovered that exhaustive search is not considered a robust searching technique, as it relies on brute force or random search mechanisms, which increase the risk of overfitting and limit variations in the searching process [22–24].

## Related studies

The development of novel metaheuristic algorithms has brought relief to the artificial neural network, artificial intelligence, and machine learning communities. These algorithms can involve the network at various stages, such as parameter estimation, system optimization, adjustment and weight training, system adaptation to determine the number of layers, node transfer functions, learning rules, and retrieval phase.

Attempts have been made by various researchers to overcome the issue of premature convergence associated with the structure of the Hopfield neural network. This includes the research conducted by [20], where a direct technique was employed to determine the existence and global exponential stability of a nearly automorphic solution for Clifford-valued high-order Hopfield neural networks (CHNN) with leakage delays. Instead of immediately examining Clifford-valued systems, the system under investigation was not divided into real-valued systems. Fresh findings were produced using the employed techniques, with examples based on the given case. Another investigation was carried out by [25]. In the study, the activation was sequenced based on geometric feature correlation in image hyperplanes to overcome convergence issues. The results indicated that the suggested model outperformed four existing filters when regularized under cohomology, allowing it to function as an unconventional filter for pixel spectral sequences.

The algorithms that have been developed for various scientific and engineering applications can be embedded into the network to enhance the learning and retrieval process. Several studies have been conducted on the application and utilization of metaheuristic algorithms in solving various optimization problems. One such study conducted in [26] focused on modified Particle Swarm Optimization (PSO) based optimization algorithms for large-scale nonlinear optimization problems. The modification of the original PSO in this study incorporated a local search technique to optimize the parameters of a fuzzy classification subsystem in a series of hybrid electric vehicles (SHEV), aiming to reduce harmful pollutant emissions. The results demonstrated that the proposed technique was simple, easily implementable, and had low computational complexity, outperforming the original PSO and the clonal selection-based artificial immune system algorithm (CLONALG).

A study conducted in [27] proposed three models (ANN, MF-ANN, GEP) to predict ground vibrations from tunnel blasting using artificial intelligence techniques. The MF-ANN model outperformed others in accuracy and efficiency, providing valuable information for safety assessments. similarly, [28] also used ANN models (including PSO-ANN) to predict the environmental impact of tunnel blasting, with the PSO-ANN model showing superior performance. These models offer accurate methods for assessing ground vibrations and the environmental effects of tunnel blasting. Another study proposed the use of the Graph Long Short-Term Memory (GLSTM) neural network and the Dragonfly algorithm for node localization [29]. This approach aimed to predict pollution levels in a wireless healthcare system that has been revolutionized by the integration of smart technologies. The GLSTM neural network provided an efficient and accurate method for pollution level prediction, while the Dragonfly algorithm accurately localized the nodes to facilitate efficient data transfer. The proposed system

has the potential to significantly improve the accuracy of pollution prediction and node localization in wireless healthcare systems [30].

Reducing energy consumption and optimizing the lifetime of wireless sensor networks are crucial objectives. However, some clustering algorithms, such as LEACH, may not deliver satisfactory performance [31]. To enhance LEACH's effectiveness, researchers have proposed an improved version of the Dragonfly algorithm for load balancing. Comparative simulations were conducted with traditional LEACH and particle swarm optimization algorithms, revealing that the enhanced Dragonfly algorithm outperformed the others. Maintaining reliable and secure routing protocols in Mobile Ad hoc Networks (MANETs) is essential due to the dynamic and open nature of wireless communication. Black hole attacks, where compromised nodes act as false routers, pose significant threats. To address this, a novel approach leveraging the Firefly Algorithm and Artificial Neural Network has been introduced to enhance the Ad hoc On-Demand Distance Vector (AODV) routing protocol [32]. Extensive numerical experiments evaluated computation overhead, packet delivery rate, throughput, and delay, demonstrating the superior performance of the proposed approach compared to traditional methods in mitigating black hole attacks in MANETs. In the context of matching engines, a heuristic algorithm has been developed to reduce memory demand while achieving effective and high-performance results [33]. This algorithm estimates distances between strings in a unique pattern, facilitating rule classification and enhancing the matching engine's capabilities

Recently, researchers proposed a novel approach that combines the Gravitational Search Algorithm (GSA) and Deep Q-Learning (DQL) algorithm using Reinforcement Learning (RL) [34]. The hybridization of GSA and DQL was utilized, with GSA initializing the weights and biases of the neural network in DQL to ensure stability. The proposed approach demonstrated superior performance compared to similar techniques. In a separate study, fuzzy controllers were employed to enhance the performance of control systems in electromagnetic-actuated clutch systems [35]. The parameters of the fuzzy controllers, including membership functions and rules, were optimized using the Grey Wolf Optimizer (GWO). The Takagi-Sugeno type-2 fuzzy controller exhibited greater efficiency in handling complex processes. A new swarm-based metaheuristic algorithm, called Tuna Swarm Optimization (TSO), was developed based on the cooperative foraging behaviour of tuna swarms. The TSO algorithm outperformed other comparative algorithms in terms of optimization performance [36]. Another intriguing study focused on the optimization capabilities of the Cat Swarm Optimization (CSO) algorithm [37]. The CSO was found to be a robust and powerful swarm-based metaheuristic optimization approach, surpassing other algorithms in solving optimization problems [38].

In the realm of algorithm development, novel metaheuristics algorithms have emerged as potential solutions to address the convergence problem of networks, facilitating faster learning and testing phases. One such approach involves the integration of a genetic algorithm (GA) with the Hopfield artificial neural network (HNN) [39]. The objective is to leverage the optimization capacity of the GA to enhance the learning process of the HNN and improve the model's overall performance. Additionally, a hybrid artificial ants colony (ACO) has been introduced in the learning phase of neural networks, specifically in the context of discrete optimization for data analysis and data mining techniques [40]. This integration aims to exploit the strengths of ACO in addressing discrete optimization problems within the neural network framework.

To address the challenging of Boolean Satisfiability (SAT) problem representation, a modified version of the Hopfield Artificial Neural Network (MHNN) has been [39]. The purpose of this approach is to assess the efficiency of the MHNN model in solving SAT, particularly in comparison to existing methods. The proposed neural network model is compared with traditional Greedy SAT and genetic algorithms (GA) for SAT. The results demonstrate that

MHNN can effectively serve as a viable alternative for solving Boolean SAT, offering favourable output quality and response time speed.

A hybrid model combining artificial immune systems (AIS) and Case-based Reasoning (CBR) was proposed to manage the processes of adaptation (reuse and revision), recovery, and retention of cases [41]. The aim was to offer an alternative approach for identifying high-density areas, clustering, enhancing search efficiency within the search space, and storing relationships among similar cases. The proposed model was applied to address the problem of fault detection and diagnosis. The obtained results were compared using specific performance metrics for CBR, revealing promising prospects for the proposed model.

An application of Ant Colony Optimization (ACO) in optimization problems for classification purposes was presented by [40]. The objective was to develop a model that captures the relationships between input attributes and the target class in a dataset. The proposed classification model aimed to predict new patterns using ACO to enhance the learning structure of feed-forward neural networks. A nonparametric Friedman test was employed to determine statistical significance, comparing the proposed model with existing evolutionary algorithms for evolving neural networks. The results demonstrated the efficiency of the proposed model in handling classification problems. Similarly, [42] conducted a study using the Imperialist Competitive Algorithm (ICA). The study aimed to compare the performance of ICA in the training and testing phases of Hopfield Neural Networks (HNN) using simulated real-life datasets against Exhaustive Search (ES) and a standalone Genetic Algorithm (GA) for 3-Satisfiability. The results of both studies indicated that incorporating ICA in the learning and training phases of HNN resulted in improved classification accuracy and lower error accumulation compared to ES and standalone GA. A novel Election Algorithm (EA) as a heuristics search technique in a Hopfield-type artificial neural network (HNN) for solving random satisfiability problems using a simulated dataset was proposed in [43]. The main objective was to assess the effectiveness of the Election Algorithm (EA) in improving the learning phase of the HNN for random k-Satisfiability logical rules. The results of the proposed HNN-RAN$k$SAT-EA model showed promising performance, demonstrating favourable agreement with the existing HNN-RAN$k$SAT-ACO approach while outperforming the traditional HNN-RAN$k$SAT-ES method.

Furthermore, a recent advancement was made in upgrading the Random 2-Satisfiability (RAND-2SAT) model, proposed by [43] to Random 3-Satisfiability (RAND-3SAT) by [44]. The purpose of this upgrade was to incorporate high-order logical rules into the Hopfield neural network and explore the feasibility of the proposed Election Algorithm in learning for high-order logic. The results revealed that the Election Algorithm demonstrated optimal performance in the Hopfield neural network when applied to high-order logic problems

The Dragonfly algorithm has emerged as an innovative evolutionary metaheuristic algorithm, effectively applied in engineering applications and computational optimization to find optimal solutions. In a study conducted by [45], a novel artificial dragonfly algorithm (ADA) was developed, inspired by the intelligent swarming behaviours observed in natural interactions of dragonflies, such as navigation, food search, and enemy avoidance. The main focus of ADA was to mimic the dynamic and static swarm behaviours exhibited by dragonflies in nature. This design allows ADA to possess efficient capabilities for both exploration (global search) and exploitation (local search), making it a simple and efficient algorithm suitable for integration into the learning or training process of any neural network model, including HNN. The convergence of ADA is guaranteed during the optimization process, as appropriate weights can be adaptively assigned to each ADA operator. This adaptive weighting facilitates a smooth transition from exploration (local search) to exploitation (global search) of the search space, ensuring the convergence of the dragonflies towards the optimal solution.

The dragonfly algorithm and its modified variants have proven to be successful in various optimization and search problems across mathematical and engineering applications. For instance, [46] proposed a memory-based version of the hybrid dragonfly algorithm (MHDA) specifically tailored for solving numerical optimization problems in engineering applications. In another study, [47] introduced a binary version of the dragonfly algorithm for feature selection. They demonstrated its effectiveness in selecting relevant features for optimization tasks. A similarly study in [46] proposed the ADA algorithm, an evolving metaheuristic algorithm, for solving the static economic dispatch problem in solar energy. ADA proved to be a useful optimization tool for constrained optimization problems in this context. Additionally, in [48] a novel approach utilizing ADA in an optimization technique to determine the optimal threshold value for image segmentation was proposed. Furthermore, [49] proposed the Dragonfly Chaotic algorithm for feature selection, showcasing the algorithm's applicability in this domain. A function optimization approach based on the ADA and Multilayer Perceptron Training in Neural Networks was developed in [50]. Their computational experiments on benchmark problems demonstrated the efficacy of ADA, particularly for multilayer perceptron training in neural networks.

In a recent study by [51], the dragonfly algorithm (ADA) was employed for the dynamic scheduling of assignments in cloud computing, aiming to find near-optimal solutions. The results obtained highlighted the performance of dragonfly metaheuristic algorithms in resource management and their potential for customization to meet specific requirements in cloud computing scenarios

In a related study conducted by [43], a novel hybrid discrete version of the artificial dragonfly algorithm (DADA) was developed for Exact Satisfiability representation using agent-based modelling (ABM). The main objective was to optimize the states of neurons within a dynamic system implemented on the NETLOGO platform. The DADA algorithm was chosen due to its capability to provide diverse solutions through random searching and a static swarm mechanism, enabling the convergence of computational problems towards the best global optimal search space. The proposed DADA algorithm was compared with a genetic algorithm (GA), and the results demonstrated its efficient performance in optimizing Exact-$k$SAT logical representations. The DADA-ABM approach shows great potential for modelling and optimizing complex networks that cannot be effectively captured by traditional optimization modelling techniques.

The Exact Boolean $k$Satisfiability (EB$k$SAT) problem, widely recognized as a challenging problem in computer science with diverse applications, continues to demand more effective and efficient solution methods. While various algorithms have been developed to tackle this problem, there is still room for improvement. The artificial dragonfly algorithm (ADA) has shown promise in solving optimization problems, including EB$k$SAT. However, the potential advantages of integrating ADA with the Hopfield neural network (HNN) to enhance performance in addressing the EB$k$SAT problem have not been extensively investigated.

In light of this, the objectives of this study are as follows:

1. To investigate the performance of the artificial dragonfly algorithm in solving the EB$k$SAT problem compared to other state-of-the-art algorithms.

2. To evaluate the effectiveness of the Hopfield neural network in enhancing the performance of the artificial dragonfly algorithm in solving the EB$k$SAT problem.

3. To determine the effect of different parameters, such as the dragonfly population size, the number of neurons in the Hopfield network, and the learning rate, on the performance of the proposed algorithm.

4. To develop an efficient and effective algorithm based on the artificial dragonfly algorithm in the Hopfield neural network for solving the EBkSAT problem.

5. To compare the performance of the proposed algorithm with other state-of-the-art algorithms on a set of benchmark instances.

## Research questions

The purpose of this study is to address the following research questions:

1. What is the performance of the artificial dragonfly algorithm in solving the EBkSAT problem compared to other state-of-the-art algorithms?

2. How effective is the artificial dragonfly algorithm in enhancing the performance of the Hopfield neural network in solving the EBkSAT problem?

3. What is the impact of different parameters, such as the size of the dragonfly population, the number of neurons in the Hopfield network, and the learning rate, on the overall performance of the proposed algorithm?

4. How can an efficient and effective algorithm based on the artificial dragonfly algorithm in the Hopfield neural network be developed to solve the EBkSAT problem?

5. How does the proposed algorithm perform compared to other state-of-the-art algorithms when tested on a set of benchmark instances?

The findings of this study will make valuable contributions to the advancement of nature-inspired algorithms for solving combinatorial optimization problems, thereby carrying practical implications across various fields. The proposed hybrid computational model presented in this paper offers an alternative approach to tackle different combinatorial optimization problems, making it particularly relevant for the fields of computational science and mathematics.

The paper is structured as follows: Section 2 outlines the methodology adopted in this study, encompassing 2.1 the formulation of Exact Boolean kSatisfiability (EBkSAT) of a Boolean Formula, 2.2 the mapping of EBkSAT in the Hopfield neural network (HNN) model, 2.3 the learning phase of HNN, and 2.4 the algorithms of the artificial dragonfly and the introduced hybrid algorithm that integrates the artificial dragonfly into the HNN to achieve optimal Satisfiability representation. In Section 3, we present the experimental setup of the model, while Section 4 presents the experimental results along with a comprehensive discussion and conclusions are drawn from this exploration

## Materials and methods

### Exact Boolean *k*Satisfiability (EB*k*SAT)

The EBkSAT problem involves a Boolean formula that represents a particular decision problem. It aims to determine whether a given Boolean satisfiability formula in Conjunctive Normal Form (CNF) holds a true representation, satisfying a specified literally in every clause. If this condition is met, it confirms the existence of a label representation. Conversely, if the condition is not satisfied, it indicates the absence of a label representation [52]. The EBkSAT problem can be seen as a variant of the Boolean Satisfiability (SAT) problem, where the input instance is similar, but with a distinction in the EBkSAT representation. In EBkSAT, a clause is considered satisfied only if exactly one of its literals is true, in contrast to the requirement of at least one literal being true in ordinary k-SAT formulations.

Let's consider a Boolean expression in which $F_{EBkSAT}$ is built from Boolean variables in CNF with the following properties.

1. Comprising a Boolean variables' set, $(x_1, x_2, x_3, \ldots, x_n)$, whereby $x_i \in \{1, -1\}$;

2. A group of literals, whereby a literal represents a given variable $x_i$ or a variable's negation $\neg x_i$;

3. A group of $m$ distinct logical clauses $C_i \in (c_1, c_2, c_3, \ldots, c_m)$;

4. Each satisfying assignment satisfies exactly one literal in each clause;

5. Each variable in the logical clause is linked by Boolean connectives OR ($\vee$);

6. Every logical clause comprises a literal connected by the Boolean connective AND ($\wedge$);

7. Each clause $C_i$ is a disjunction of exactly three literal and contains at most three literals.

These properties simplify the formulation of the problem via a Hopfield neural network (HNN) and preserve its NP-completeness [53]. The general formulation of EBkSAT is presented as follows.

$$F_{EBkSAT} = \bigwedge_{i=1}^{k} C_i \tag{1}$$

when $k = (1,2,3)$ in Eq (1) describes the Boolean formula for EBkSAT containing logical clause $C_i$ given in Eq (2) as follows:

$$C_i = \bigvee_{j=1}^{3} (E_{ij}, D_{ij}, F_{ij}) \tag{2}$$

If the context is clear, we denote the number of clauses as EBkSAT [54]. The Boolean values are presented in bipolar $E_{ij}, D_{ij}, F_{ij} \in [1, -1]$ representing the TRUE value of a mapping or its FALSIFICATION respectively. Examples of EBkSAT formulation when $k = 3$ is presented as follows:

$$F_{EBkSAT} = (E_1 \vee E_2 \vee E_3) \wedge (D_1 \vee \neg D_2 \vee D_3) \wedge (\neg F_1 \vee F_2 \vee F_3) \tag{3}$$

Eq (3) is satisfiable since it gives true value, resulting in Eq (4).

$$F_{EBkSAT} = 1 \tag{4}$$

If the neuron states are considered as $E_i (i = 1,2,3)$, $D_i (i = 1,2,3)$ and $F_i (i = 1,2,3)$ then the Boolean expression will be unsatisfiable only if

$$F_{EBkSAT} = -1 \tag{5}$$

In this study, $F_{EBkSAT}$ has been embedded into HNN in as a proposed model, EBk-SAT-DHNN in the next section. To the knowledge of the author, an artificial dragonfly algorithm has not been applied before in accelerating the HNN learning phase to attack the Exact $k$satisfiability representation.

## Mapping Exact of the Boolean *k*Satisfiability in the Hopfield neural networks

The basic structure of the Hopfield neural network (HNN) consists of several components, including inputs, outputs, and weights. In a stable HNN, the energy decreases over time. The network continuously converges towards a fixed point, which represents stable neural states.

This property makes HNN suitable for solving various search and optimization problems [55]. The energy of a stable HNN decreases over time. The fundamental architecture and structure of a discrete HNN with $n$ neurons can be defined by two $nxn$ real matrices.

$$R^0 = (r_{ij}^0)_{n \times n}, R^1 = (r_{ij}^1)_{n \times n} \tag{6}$$

an $n$-dimensional vector is presented as follows

$$\tau = (\tau_1, \tau_2, \ldots, \tau_N)^T \tag{7}$$

where N is denoted by

$$N = (R^0 \oplus R^1, \tau) \tag{8}$$

The state of each neuron has been denoted by two possible values of 1 or -1 of neuron $i$ at a time as follows,

$$t \in \{0, 1, 2, 3, \ldots\} \tag{9}$$

$s_j(t)$ is the initial input vector pattern presented to the network as follows.

$$S_j(t) = (s_1(t), s_2(t), \ldots s_n(t))^T \tag{10}$$

Eq (10) represents the whole neurons' state at time $t$, and the states set is $S_j$ I defined as follows

$$S_j \in [-1, 1], S_j = \{(s_1, s_2, \ldots s_n)^T | s_j = -1 \ or \ 1, 1 \leq j \leq n)\} \tag{11}$$

The neurons in HNN can be represented in binary $S_j \in [0,1]$ or bipolar form $S_j \in [-1,1]$ based on the neurons dynamics as follows,

$$S_j(t) \rightarrow \text{sgn}(h_j(t)) \tag{12}$$

Eq (12) is defined by the *Ising* variables in the spin-glass and Dean's mechanical physics problem [56], whereby the local field, $h_j(t)$ is described as follows:

$$h_j(t) = \sum_{j=1, i \neq j}^{m} \lambda_{ij}^{(2)} S_j(t) + J_i^{(1)} \tag{13}$$

The benefit of utilizing the bipolar values rather than the binary values involves the network's symmetry of states. If a given pattern $S_j$ in a bipolar form can be stable, then its inverse can be stable as well. However, the general asynchronous updating state of the discrete HNN model is discrete in time, performing as follows:

$$S_j(t+1) = \begin{cases} 1, & if \ \sum_{j}^{N} \lambda_{ij} S_j(t) + \tau_j \\ -1, & otherwise \end{cases} \tag{14}$$

whereby $\lambda_{jk}$ represents a synaptic connection matrix, which determined a strong connection between the $j$ and the $k$ neurons, $S_j$ refers to this unit condition $k$, and $\tau_j$ describes the neuron's threshold function $j$. Some studies, including the work of [57], and [23] defined $\tau_j = 0$ verify that the energy state of HNN network often decreases monotonically and every time the neuron is linked to $\lambda_{jk}$, a synaptic connection value is preserved in a form of a stored pattern in the interconnected vector, whereby the vectors $\lambda^{(1)} = [\lambda_{jk}^{(1)}]_{n \times n}, \lambda^{(2)} = [\lambda_{jk}^{(2)}]_{n \times n} \lambda^{(3)} = [\lambda_{ijk}^{(3)}]_{n \times n}$ or N-

dimensional variable vectors presented in Eq (7). The synaptic weight matrix constraint $\lambda^{(1)}$, thereby not allowing a self-loop connection of the neurons as follows,

$$\lambda_{jjj}^{(3)} = \lambda_{kkk}^{(3)}, \ldots, = \lambda_{iii}^{(3)} = 0 \tag{15}$$

and symmetrical neuron synaptic weight matrix presented as follows

$$\lambda_{kji}^{(3)} = \lambda_{ijk}^{(3)} = \lambda_{jik}^{(3)}. \tag{16}$$

The energy dynamics function of the HNN model with CAM provides a versatile and high-capacity system with error tolerance and fast memory recovery capabilities, even with partial inputs [58] and [59]. By leveraging the HNN network as a logical rule, it becomes suitable for integration into combinatorial optimization problems like SAT. This involves assigning neurons to each variable in EBkSAT using specific cost functions and a generalized cost function, thereby instructing the system configuration's performance based on the synaptic strength matri $E_{F_{EBkSAT}}$, which is controlling the neurons' combinations in the network and $F_{EBkSAT}$ is as follows,

$$E_{F_{EBkSAT}} = \sum_{i=1}^{NN} \prod_{j=1}^{N} T_{ijk} \tag{17}$$

Whereby $N$ as well as $NN$ represent the number of the variables and the given number of the generated neurons in $F_{EBkSAT}$ correspondingly and the inconsistency of $F_{EBkSAT}$ representation is defined as follows:

$$T_{ij} = \begin{cases} \dfrac{1}{2}\left(1 - S_\rho\right), & if \; \neg\rho \\ \dfrac{1}{2}\left(1 + S_\rho\right), & otherwise \end{cases} \tag{18}$$

This value $E_{F_{EBkSAT}}$ can be proportional to a number of "inconsistencies" of clauses ($E_{ij} = -1$, $D_{ij} = -1$, $F_{ij} = -1$). The minimum $E_{F_{EBkSAT}}$ parallels the "most consistent" selection $S_j$. Consequently, an updating state $E_{F_{EBkSAT}}$ in the Hopfield neural network in the given Eqs (12) and (13) is upgraded to obey a third-order connection, which is described in Eqs (19) and (20) correspondingly as follows:

$$h_j(t) = \sum_{i=1, i \neq j, j \neq k}^{m} \sum_{i=1, i \neq j, j \neq k}^{m} \lambda_{ijk}^{(3)} S_j(t) + \sum_{i=1, i \neq j, j \neq k}^{m} \lambda_{ij}^{(2)} S_j(t) + \lambda_i^{(1)} \tag{19}$$

In such a case, two output values for each neuron can be possible as follows:

$$S_j(t+1) = \begin{cases} 1, & \displaystyle\sum_{i=1, i \neq j, j \neq k}^{m} \sum_{i=1, i \neq j, j \neq k}^{m} \lambda_{ijk}^{(3)} S_j(t) + \sum_{i=1, i \neq j, j \neq k}^{m} \lambda_{ij}^{(2)} S_j(t) + \lambda_i^{(1)} \geq 0 \\ -1, & \displaystyle\sum_{i=1, i \neq j, j \neq k}^{m} \sum_{i=1, i \neq j, j \neq k}^{m} \lambda_{ijk}^{(3)} S_j(t) + \sum_{i=1, i \neq j, j \neq k}^{m} \lambda_{ij}^{(2)} S_j(t) + \lambda_i^{(1)} < 0 \end{cases} \tag{20}$$

whereby $\lambda_{ijk}^{(3)}, \lambda_{ij}^{(2)}$, as well as $\lambda_i^{(1)}$ are a third, a second, and an initial order synaptic weight for embedded into $F_{EBkSAT}$. Eqs (18) and (19) guarantee that neurons $S_j$ always converge $E_{F_{EBkSAT}} \to 0$. Thus, the given Lyapunov energy function (LEF) $H_{F_{EBkSAT}}$ was employed to ensure the network's energy dynamics states decrease monotonically. The retrieval neural states'

quality is measured via the LEF, $H_{F_{EBkSAT}}$, $k = 2$, which is given in Eq (21).

$$H_{F_{EBkSAT}} = -\frac{1}{2}\sum_{i=1,i\neq j}^{m}\sum_{j=1,i\neq j}^{m}\lambda_{ij}^{(2)}S_iS_j - \sum_{i=1,i\neq j}^{m}\lambda_i^{(1)}S_j \tag{21}$$

The energy state always change to a negative state until the global minimum energy is reached by the system. Eq (22) represents a monotonic drop with dynamics and is possibly upgraded to incorporate the third-order connections as follows,

$$H_{F_{EBkSAT}} = -\frac{1}{3}\sum_{i=1}^{m}\sum_{j\neq k}^{m}\sum_{k=1,i\neq k}^{m}\lambda_{ijk}^{(3)}S_iS_jS_k - \frac{1}{2}\sum_{i=1,i\neq j}^{m}\sum_{j=1}^{m}\lambda_{ij}^{(2)}S_iS_j - \sum_{i=1,i\neq j}^{m}\lambda_i^{(1)}S_j \tag{22}$$

The network will be generating the needed solution if the induced neuron state reaches global minimum energy (an equilibrium state). The Energy state of Eqs (22) and (23) always portrayed $F_{EBkSAT}$ decreases monotonically to a certain configuration (zero). Thus, the value $H_{F_{EBkSAT}}$ demonstrates this energy value with the total final energy $H_{F_{EBkSAT}}^{\min}$ reached $F_{EBkSAT}$. Therefore, as the network approaches the final energy state, the network energy changes approach zero. The quality of the ultimate neuron state can be maintained according to Eq (24) as follows.

$$|H_{F_{EBkSAT}} - H_{F_{EBkSAT}}^{\min}| \leq \xi \tag{23}$$

where the parameter $\xi$ refers to the value of the pre-determined tolerance. The value parameter $\xi = 0.001$ has been taken as a tolerance value in [57] and [60]. However, other tolerance values can be considered.

If the $F_{EBkSAT}$ logical representation embedded in HNN does not satisfy the criteria state in Eq (24), then the ultimate state achieved has been trapped in a wrong pattern (a local minimum resolution). Fig 1 Shows a flowchart of HNN learning process based on Wan Abdullah learning approach while in Table 1 is the HNN algorithm.

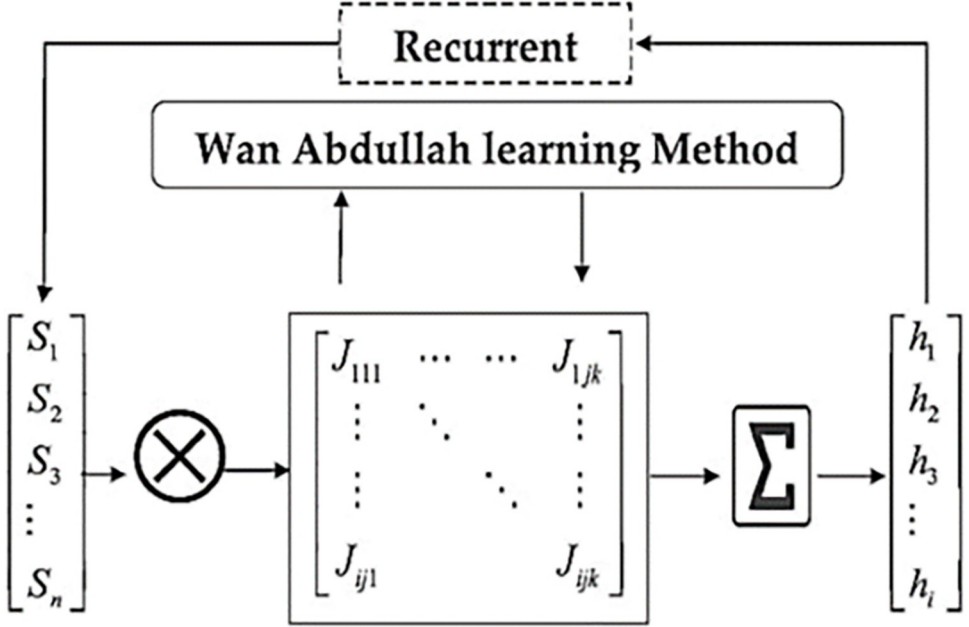

**Fig 1. Flow diagram of Wan Abdullah method for HNN.**

**Table 1. The Hopfield artificial neural network algorithm.**

```
Algorithm 1 The algorithm of the Hopfield artificial neural network
Step1: Set Sⱼ(t) = (s₁(t),s₂(t),…sₙ₋₁(t),sₙ(t))ᵀ the inputs pattern and store it as CAM
Step2: Initialize the weights of synaptic matrices λ⁽³⁾ᵢⱼₖ,λ⁽²⁾ᵢⱼ,λ⁽¹⁾ᵢ,λ⁽¹⁾ⱼ, and λ⁽¹⁾ₖ
Step3: Compute HNN output vectors, hⱼ(t) = (h₁(t),h₂(t),h₃(t),…hₙ₋₁(t),hₙ(t))
Step4: Compute changes of HNN Lyapunov energy state H_{F_{EBkSAT}}
Step5: If the condition stated in Eq. (23), is not satisfied, go to step 2, and
update HNN output vectors, Update the synaptic weights matrices by Wan
Abdullahi learning or Hebbian Learning.
Step6: If the condition stated in Eq. (23) is satisfied, proceed to step 7
Step7: Initialize the EBkSAT logic program and assign neurons to each variables
Step8: If EBkSAT is not satisfied, go to step 2, otherwise proceed to step 9.
Step9: Compute the fitness values of the EBkSAT using Eq. (18).
If fitness criteria in Eq. (24) are satisfied, then end the program and print
solutions, otherwise, go to step 2.
```

For the HNN algorithm in Table 1, in case the fitness requirement is not met, the program will continue to operate in an iterative manner. Conversely, if the solution meets the requirement, it will be terminated and printed. The HNN's fitness requirement is met if the output of the network corresponds to a specific constant value, which indicates that no further searching/optimization is necessary in the objective function. Additionally, all decision variables must be non-negative, and no constraints should be violated. Until now, no research has combined an ADA discrete version with an HNN discrete version as a single computational model. Consequently, the ADA robustness contributes to enhancing the HNN training process.

## 2.1 Learning phase in Hopfield artificial neural network for Exact Boolean *k*Satisfiability

The learning phase of the HNN system is introduced by inserting the "behaviour" of the EB*k*-SAT in the HNN. The behaviour of EB*k*SAT can be implemented by searching the correct synaptic weight vector of EB*k*SAT logical representation. Finding the correct weight vector for the EB*k*SAT clause is the primary objective of the HNN-EB*k*SAT learning phase. The implementation of $F_{EBkSAT}$ in the HNN in this work is designated as the HNN-EB*k*SAT model. However, there was no previous attempt to implement the EB*k*SAT logical representation in HNN.

Consider the following EB*k*SAT program:

$$F_{EBkSAT} = E_1, E_2 E_3 \leftarrow$$
$$F_2, F_3 \leftarrow F_1 \tag{24}$$
$$D_1, D_3 \leftarrow D_2$$

Given the goal of the program

$$\leftarrow F \tag{25}$$

where *F* is the conjunction of clauses, which defined the goal of the logic program, whose task involves showing that ←*F* is incompatible to confirm the goal *F*. This is represented as a combinatorial optimization problem, whereby the "inconsistency" of Eq (25) is to be optimized (Minimization problem). The inconsistency is presented as a negation of Eq (25) after translating all clauses into Boolean algebraic form and negating as follows,

$$\neg F_{EBkSAT} = (\neg E_1 \lor \neg E_2 \lor \neg E_3) \land (\neg D_1 \lor D_2 \lor \neg D_3) \land (F_1 \lor \neg F_2 \lor F_3) \tag{26}$$

The costs function of Eq (26) is the Boolean algebra formula for bipolar representation represents the $F_{EBkSAT}$ logical inconsistency, following Wan Abdullah's learning method, which is given herein:

$$E_{F_{EBkSAT}} = \frac{1}{2}\left(1 - S_{E_1}\right)\frac{1}{2}\left(1 - S_{E_2}\right)\frac{1}{2}\left(1 - S_{E_3}\right) + \frac{1}{2}\left(1 - S_{D_1}\right)\frac{1}{2}\left(1 + S_{D_2}\right)\frac{1}{2}\left(1 - S_{D_3}\right)$$
$$+ \frac{1}{2}\left(1 + S_{F_1}\right)\frac{1}{2}\left(1 - S_{F_2}\right)\frac{1}{2}\left(1 - S_{F_3}\right) \tag{27}$$

where ($i$ = 1,2,3) represent the truth values of neurons $E_i, D_i \& F_i$ respectively. In this work, $S_{E_i}, S_{D_i}, \& S_{F_i}$ can take any of the two possible values of 1 (True) and -1(False) for deriving the cost function to be minimized $E_{F_{EBkSAT}}$. The optimum value $E_{F_{EBkSAT}} \to 0$ refers to the entire satisfied logical clauses. This value $E_{F_{EBkSAT}}$ is related to the unsatisfied logical clauses number [57] and [60]. It can be applied in the given network via the storing atom truth values, thereby producing an optimized cost function in case optimal clauses are signified. The cost function $E_{F_{EBkSAT}}$ in Eq (17), when programmed onto a third order logic, the energy $H_{F_{EBkSAT}}$ in Eq (23) yields the correct synaptic strengths of HNN-EBkSAT to be stored as CAM of HNN. These synaptic weights are, thus, utilized throughout a recovery phase. Thus, the training process can provide an optimum cost function for determining the optimum synaptic weights. Eq (23) can further be expanded and simplified by considering all neurons connections associated with $H_{EBkSAT}(k = 3)$,

$$E_{F_{EBkSAT}} = -\frac{1}{3}\left(6J^{(3)}_{[E_1E_2E_3]}S_{E_1}S_{E_2}S_{E_3} + 6J^{(3)}_{[D_1D_2D_3]}S_{D_1}S_{D_2}S_{D_3} + 6J^{(3)}_{[F_1F_2F_3]}S_{F_1}S_{F_2}S_{F_3}\right) - \frac{1}{2}\left(2J^{(2)}_{[E_1E_2]}S_{E_1}S_{E_2} + 2J^{(2)}_{E_1E_3}S_{E_1}S_{E_3}\right.$$
$$+ 2J^{(2)}_{[E_2E_3]}S_{E_2}S_{E_3} + 2J^{(2)}_{[D_1D_2]}S_{D_1}S_{D_2} + 2J^{(2)}_{[D_2D_3]}S_{D_2}S_{D_3} + 2J^{(2)}_{[D_1D_3]}S_{D_1}S_{D_3} + 2J^{(2)}_{[F_1F_2]}S_{F_1}S_{F_2} + 2J^{(2)}_{[F_1F_3]}S_{F_1}S_{F_3}$$
$$\left. + 2J^{(2)}_{[F_2F_3]}S_{F_2}S_{F_3}\right) - \left(J^{(1)}_{E_1}S_{E_1} + J^{(1)}_{E_2}S_{E_2} + J^{(1)}_{E_3}S_{E_3} + J^{(1)}_{D_1}S_{D_1} + J^{(1)}_{D_2}S_{D_2} + J^{(1)}_{D_3}S_{D_3} + J^{(1)}_{F_1}S_{F_1} + J^{(1)}_{F_2}S_{F_2} + J^{(1)}_{F_3}S_{F_3}\right) \tag{28}$$

The improved global minimum energy is described as the projected global minimum energy to be achieved when a retrieval process ends. An EBkSAT logic program represents a combinatorial optimization problem and, therefore, the improved global minimum energy will be calculated in Eq (20). According to Eq (26), $F_{EBkSAT}$ consists of 3 logical clauses with 9 randomly selected variables from a pre-determined 9 set of variables, 7 positive laterals, and 2 negative literals,

$$S_{E_1} = S_{E_2} = S_{D_1} = S_{F_3} = S_{D_3} = S_{E_3} = S_{D_2} = 1; S_{F_1} = S_{F_2} = -1 \tag{29}$$

Eq (29) is one of the consistent interpretations that make the entire Boolean formula $F_{EBkSAT}$ true. Substituting Eq (28) into Eq (28), the expected global minimum energy is obtained as in Eq (30).

$$H^{\min}_{F_{EBkSAT}} = -\frac{3}{8}. \tag{30}$$

$H^{\min}_{F_{EBkSAT}}$ is used for separating the neuron state correctness, which is produced by this network throughout the retrieval phase.

## 2.2 Proposed artificial dragonflies algorithm in HNN for EBkSAT representation

The objective of EBkSAT is to decide whether a given Boolean formula in the CNF or Conjunctive Normal Form has a given truth assignment, which correctly satisfies *one* literal in every clause or determines the absence of label assignment. Since the original dragonfly

algorithm was developed to address optimization problems incorporating continuous function, the SAT problem is a given discrete optimization problem. ADA in this work has been adapted to deal with EBkSAT logic. These steps demonstrate the applied procedure in the suggested approach. One of the objectives is the application, which involves the ADA operators' reformulation to tackle discrete optimization instead of continuous optimization.

The given limitation necessitates an algorithm, which effectively flips this neuron state following the former enhanced solution using a wide space of solution. The value of the fitness function of a candidate $d_r^f$ in the ADA is provided in Eq (31).

$$\lambda_{F_{EBkSAT}} = d_1^f(x) + d_2^f(x) + d_3^f(x) +, \ldots, d_{N_{Dragonfly}}^f(x) = \sum_{i=0}^{m} C_i^{(k)} \tag{31}$$

where $f \in [1,2,3,\ldots N]$ and $d_r^f$ refers to the location of $r$th dragonfly in $d$th dimensional space, $N_{dragonfly}$ denotes the prospective search agents' number. Where $\lambda_{F_{EBkSAT}}$ refers to the satisfied clauses' maximum number, $m$ describes the clause's maximum number in $F_{EBkSAT}$ the logical program and $C_i^{(k)}$ denotes the tested clauses by using ADA.

$$C_i^{(k)} = \begin{cases} 1 & , \; Satisfied \\ 0 & , \; otherwise \end{cases} \tag{32}$$

Each of the neuron strings in the HNN network refers to an assignment, which matches the Exact $k$SAT instances. The suggested ADA objective function involves maximizing the artificial dragonflies' fitness $d_r^f$ (the neuron string). In general, global optimization can be presented in Eq (33) without the loss of generality in the minimization problem.

$$Max[\lambda_{F_{EBkSAT}} = \sum_{i=0}^{m} C_i^{(k)}] \tag{33}$$

The mapping of ADA in HNN for EBkSAT as it is shortened to ADA-HNN-EBkSAT and the stages of ADA-HNN-EBkSAT are described herein:

Stage 1: *Initialization*

Any optimization or search mainly aims to find the best solution regarding the problem's the variables and vector variables to be optimized can be formed. The first population of the given size $N_{dragonfly}$ for artificial dragonflies $d_r^f = [d_1^f, d_2^f, d_3^f, \ldots, d_{N_{Dragonflies}}^f]^T$ for each solution and step matrix $\Delta d_r^f$ was generated. The state of each Artificial Dragonflies $d_r^f$ in the search space is denoted by 1 or -1 which represents the *True* or *Falsification* that corresponds to the possible mapping for the Exact $k$Satisfiability problem. Each of the solutions can be randomized within the boundaries of the variable in Eq (34) as follows.

$$d_r^f = \lambda_{d_r^f}^{\min} + d_1^f * (\lambda_{d_r^f}^{\max} - \lambda_{d_r^f}^{\min}) \tag{34}$$

where $r \in [1,2,3 \ldots N_{dragonfly}]$ and $d_r^f$ specifies a location for $r$th dragonfly in the given $f$th dimensional space, whereas $N_{dragonfly}$ specifies a prospective search agents' number. The homogeneously distributed random can be defined as $d_1^f \in [0,1]$, whereby this problem aims to find the ideal EBkSAT.

**Step 2. *Calculate the distance for each artificial dragonfly.*** The distance from the neighbourhood can be determined by calculating and selecting the Euclidean distance between the entire dragonflies and picking $N$ out of them. The distance $\sigma_{ij}$ aims to obey the Euclidean

distance metric in Eq (35) as follows.

$$\sigma_{ij} = dist(d^f_{ir}, d^f_{jr}) \tag{35}$$

where $d^f_{ir}$ and $d^f_{jr}$ are the fitness of the neighbourhood areas and artificial dragonflies, respectively. The artificial dragonfly will be assigned to its neighbourhood areas based on its fitness values.

**Stage 3: *Fitness Evaluation.*** This variable vector can be examined according to this fitness $\lambda_{F_{EBkSAT}}$ to a given quantified variable position within a modified solution space. The entire randomized variable vectors can undergo a fitness function assessment. The fitness $\lambda_{d^f_r}$ of each $d^f_r$ is computed based on the initial position variables that are generated randomly between the variables' lower and upper limits by using Eq (30) to Eq (31). The parameters of each dragonfly are the same as the variables in the optimization problem. The combination of parameters determines the attractiveness of the artificial dragonfly. In this case, the proposed model obeys $\lambda_{d^f_r} \in N_{dragonfly}$. The maximum fitness of the $d^f_r$ is given by

$$\lambda^{\max}_{d^f_r} = \lambda^{\min}_{d^f_r} \tag{36}$$

and if $\lambda_{d^f_r}$ the algorithm will be terminated when the optimum fitness is reached.

**Stage 4: *Update the position and velocity of artificial dragonflies.*** Both the position, as well as the velocity of the dragonflies' coefficients $S^f_{d_r}$, $A^f_{d_r}$ and $C^f_{d_r}$ are determined and updated (flip) based on the Eqs (37) to (39);

**Separation strategy $(S^f_r)$.** This is the states of avoidance of static collisions between artificial dragonflies in the same neighbourhood.

$$S^f_{d_r} = -\sum_{d_r=1}^{N} d^f_r - d^f_i \tag{37}$$

**Alignment strategy $(A^f_r)$.** This is the pace at which artificial dragonflies play in the same neighbourhood.

$$A^f_{d_r} = \frac{1}{N}\sum_{d_r=1}^{N} \Delta\Psi^f_i \tag{38}$$

**Cohesion strategy $(C^f_r)$.** This technique demonstrates the artificial dragonfly's inclination toward the middle of the neighbourhood. To maintain the artificial dragonflies' unity in setting the given path toward the neighbourhood core.

$$C^f_{d_r} = \left(\frac{1}{N}\sum_{d_r=1}^{N} d^f_r\right) - d^f_i \tag{39}$$

where $d^f_r$ and $\Psi^f_r$ describe the position and velocity of the $r$th potential dragonfly and $d^f_i$ described the position of this artificial dragonfly, whereas $N$ is a number of the neighbouring potential search agent.

**Stage 5: *Update Food source and Enemy source.*** The attractiveness of the artificial dragonfly toward food source and the distraction from its enemy obeys the following

equations:

$$F_{d_r}^f = \zeta_{food} - d_i^f \tag{40}$$

$$E_{d_r}^f = \zeta_{ememy} + d_i^f \tag{41}$$

where $d_i^f$ describes this artificial dragonfly position. $\zeta_{food}$ and $\zeta_{ememy}$ describe the corresponding food source, together with an enemy source. The artificial dragonfly sources for the food and the enemy are conveyed as the finest and worst solutions, correspondingly, which are so far observed in the solution space.

The adaptive integration of the preceding operations helps to correctly flip the artificial dragonfly position. This applies a phase vector and position vector to swap the artificial dragonflies' position in the search space and simulate their associated hunting movements every single round. The phase vector indicates the swapping direction of the artificial dragonflies as follows.

$$\Delta d_r^f = (sS_{d_r}^f(t) + a_r A_{d_r}^f(t) + c_r C_{d_r}^f(t) + f_r F_{d_r}^f(t) + e_r E_{d_r}^f(t)) + w_r \Delta d_r^f(t) \tag{42}$$

where $w_r$ defined the *inertia weight* of potential search agent. $s_r$, $a_r$, $c_r$, $f_r$, and $e_r$ are the separation weights, the alignment, the cohesion, the attraction of food, and the distraction of enemy, correspondingly.

In the discrete domain, the position of the artificial dragonflies can be updated by obeying the transfer function in which the velocity values are obtained as inputs and the number $d_1^f$ which represents the likelihood of switching artificial dragonfly positions in the solution space [61].

In the ADA it is seen that an artificial dragonfly moves by flipping the bits' number. Therefore, the artificial dragonfly's velocity might be represented by altering the bit probabilities that are changed for each iteration, i.e., the artificial dragonfly can move within the search space by presuming -1 or 1 values only, where every velocity is a probability of the position bit, taking the value 1. To be within the range [0,1], the velocity, which is a probability, must be limited [62]. The given function, which does this is referred to as the sigmoid function. It can be mathematically formulated as follows:

$$T(d_r^f) = \frac{1}{1 + \exp(-d_r^f)} \tag{43}$$

After calculating the probability of changing the position for the entire dragonflies, Eq (43) is used for updating (i.e., flip the neurons) the position of the search agent in these bipolar search spaces. The location switch is described by comparing it with equivalently generated random numbers between 0 and 1; these can be formulated herein:

$$d_r^f(r+1) = \begin{cases} 1 & rand[0,1] > T(d_{r+1}^f) \\ -1 & otherwise \end{cases} \tag{44}$$

with the above components, the ADA can repeatedly flip the bipolar bits in solutions until the satisfaction of an end condition.

**Stage 6: *New solution generation.*** Finally, the emergence of a new population of bipolar artificial dragonflies. Go back to Phase 2, quantify the cost of each artificial dragonfly within the new population and then perform the loop until the stop condition is met. After that, the calculated fitness function obeys both the updated location, as well as velocities. The

**Table 2. Pseudocode of artificial dragonfly algorithm.**

```
Algorithm 2 Pseudocode of artificial dragonfly algorithm
Initialize1000 population of artificial dragonfly algorithm dᶠᵣ =
[d₁ᶠ, d₂ᶠ, d₃ᶠ, ...., d_Ndragonfly]ᵀ
Initialize step vectors (Δdᶠᵣ), the number of neighbourhood (N) & maximum
iteration (λᵐᵃˣ_dᶠᵣ)
  while the end condition is unsatisfied
    Compute the Euclidean distance of every artificial dragonfly utilizing Eq.
(25)
    Estimate the fitness values of every artificial dragonfly employing Eq. (23)
    Determine the food source and enemy of the artificial dragonfly
    Modify wᵣ, aᵣ, cᵣ, fᵣ, and eᵣ
    Calculate Sᶠ_dᵣ, Aᶠ_dᵣ, Cᶠ_dᵣ, Fᶠ_dᵣ and Eᶠ_dᵣ use Eq (37) to Eq (41)
    Update neighbouring radius artificial dragonfly
    if an artificial dragonfly has at least one neighbour
    Calculate the transition probability utilizing Eq. (43)
  Update the position vector utilizing Eq. (44)
    else
Then update the artificial dragonfly's position vector employing Eq. (42)
    end if
Calculate and update the artificial dragonfly's new positions according to the
variables' boundaries
end while
```

modification of the artificial dragonfly location continues until the criteria are met. Table 2, presented the Pseudocode of ADA.

## Implementation procedure

The implementation procedure of Neuro-Heuristic searching for EBkSAT in the Hopefield Artificial Neural Network. The primary task of the program involves finding the optimum "model", which can find the optimum EBkSAT occurrences. The two variables and the clauses were randomized initially based on EBkSAT logical principles. Simulations were completed with a changing number of the neurons' complexity, i.e. $10 \leq NN \leq 120$. These models' executions, which were conducted on the EBkSAT logical representation, are presented via these steps:

Present a given logic program in Eq (25):

Translate the entire EBkSAT logical clauses to a Boolean algebra form as in Eq (26).

Define a neuron to every variable in the representation of EBkSAT logical rule Eq (27).

Randomize the neurons' state, then initialize the entire connection strengths to zero as follows.

$$J_{ijk}^{(3)} = J_{kji}^{(3)} = J_{jik}^{(3)} = 0$$

$$J_{ji}^{(2)} = J_{ij}^{(2)} = 0$$

$$J_i^{(1)} = J_j^{(1)} = 0 \tag{45}$$

Obtain the given cost function, $E_{F_{EBkSAT}}$ for the EBkSAT by utilizing Eq (27).

Compare the given cost function in Eq (26) with energy dynamics in Eq (23), then obtain the values of the synaptic weight vector as follows,

$$J_{ijk}^{(3)}, J_{ij}^{(2)}, J_i^{(1)}, J_j^{(1)}, J_k^{(1)} = 0. \tag{46}$$

Check the clause satisfaction by employing the ADA, ABC, ES, as well as AIS searching procedures, which correspond to the $E_{F_{EBkSAT}} = 0$. Then, a satisfying assignment is stored as the CAM in the HNN.

Randomize the neurons' state, then calculate the given respective local field $h_i(t)$ for a state space by utilizing Eq (20). Then, it signifies a stable configuration when it stays unchanged following five loops.

Find the network corresponding to the final state $H_{F_{EBkSAT}}$ by applying the Lyapunov energy dynamics Eq (23).

Check whether the final energy derived is a global or a local minimum based on the condition in Eq (24). Fig 2, displays the flowchart of the implementation Procedure of different algorithms in HNN with EBkSAT.

## Model experimental setup

In this study, the ADA algorithm was integrated into the HNN to search for the optimal solution for the EBkSAT model logic representation. This hybrid computational model was evaluated using three existing models from the literature. The HNN models utilized simulated datasets to establish the EBkSAT logical clauses. To ensure a meaningful comparison among the HNN models, the entire source code was developed as a simulation program using Dev C++ release version 5.11. The program was executed on a Windows 8.1 device with an Intel® Celeron® CPU B800@4GHz processor and 8 GB RAM. Table 3 provides a list of the appropriate parameters used during the execution of the ADA in the HNN model.

### Performance evaluation measure

Performance measures play a crucial role in the design process of HNN models. These measures, known as "difference measures," quantify the disparities between expected and observed values, providing a reliable assessment of the model's precision and accuracy. After the training process is successfully executed, the neural network can calculate various metrics including GmR, RMSE, MAPE, CT, and accuracy. The equations for these metrics are presented in Eqs (47) to (51).

$$TRAINING\_GmR = \frac{1}{n} \sum_{i=1}^{NN} H_{F_{EBkSAT}} \tag{47}$$

$$TRAINING\_RMSE = \frac{1}{n} \sum_{i=1}^{NN} \sqrt{\left[H_{F_{EBkSAT}}^{\min} - H_{F_{EBkSAT}}\right]^2} \tag{48}$$

$$TRAINING\_MAPE = \sum_{i=1}^{NN} \frac{100}{n} \frac{\left|H_{F_{EBkSAT}}^{\min} - H_{F_{EBkSAT}}\right|}{\left|H_{F_{EBkSAT}}^{\min}\right|} \tag{49}$$

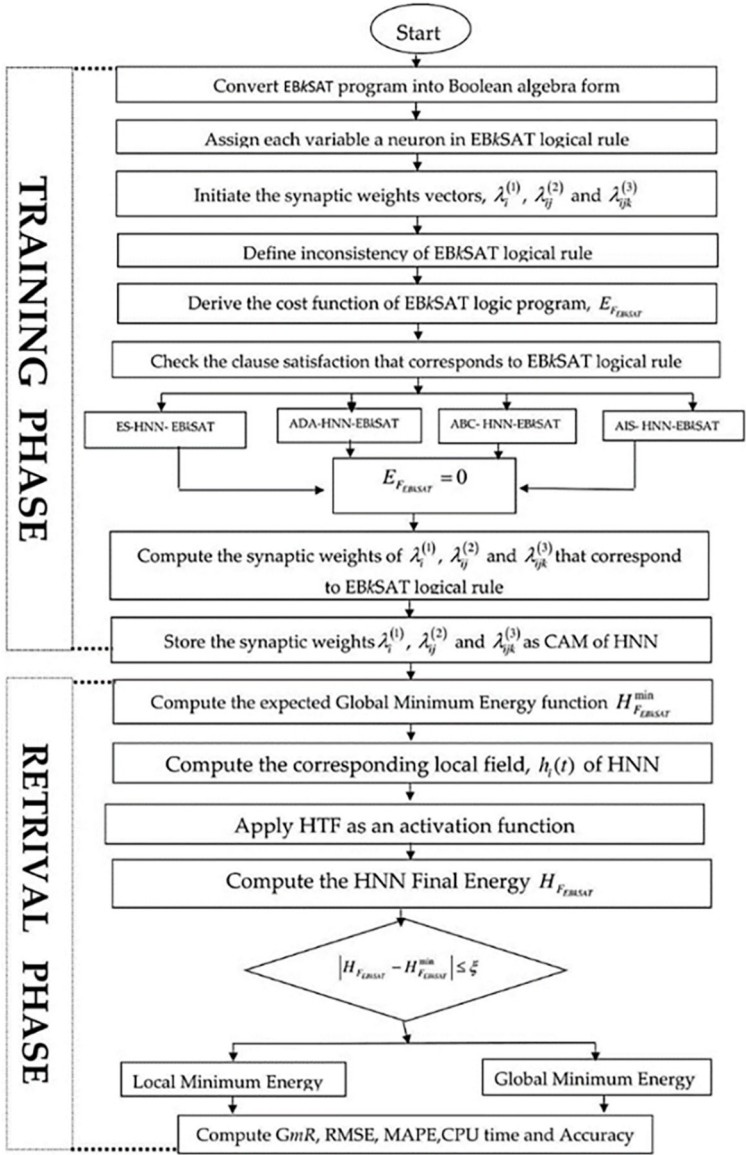

**Fig 2. Flowchart for HNN-EBkSAT implementation procedure.**

**Table 3. List of some parameters of the ADA-HNN- EBkSAT model.**

| Parameter | Value |
|---|---|
| $S_r^f$ | 0.1 |
| $A_r^f$ | 0.1 |
| $C_r^f$ | 0.7 |
| $F_{d_r}^f$ | 1.0 |
| $E_{d_r}^f$ | 1.0 |
| $N$ | 6.0 |
| $\lambda_{d_r^f}^{\max}$ | 10,000 |
| $N_{dragonfly}$ | 100 |

$$\text{TRAINING\_Time}(s) = Training\_Time(s) + \text{Recovery\_Time}(s) \tag{50}$$

$$TRAINING\_ACCURACY = \frac{P_{induced}^{Correct}}{N_{P_{test}}} \times 100\% \tag{51}$$

## Experimental results and discussion

The Artificial Dragonfly Algorithm (ADA) has been incorporated into the Hopfield Neural Network (HNN) for Exact Boolean $k$-Satisfiability Logical Representation (EB$k$SAT). The purpose is to accelerate the training capacity of the HNN for optimal representation of EB$k$SAT logical rules and to address the premature convergence behaviour of the HNN. The performance of ADA in enhancing the training process of HNN has been compared to the Artificial Bee Colony (ABC) in the Hopfield Neural Network (ABC-HNN-EB$k$SAT), the Artificial Immune System (AIS) in the Hopfield Neural Network for Exact Boolean $k$-Satisfiability Logical Representation (AIS-HNN-EB$k$SAT), and the traditional exhaustive search techniques (ES) in the Hopfield Neural Network for Exact Boolean $k$-Satisfiability Logical Representation (ADA-HNN-EB$k$SAT). Figs 3–6 show the searching capacity between the existing models and the proposed models in finding the global optimum based on stated performance metrics measures.

Fig 3 illustrates the searching behaviour of the HNN-EB$k$SAT logic representation in terms of the model's global minimum ratio (G$m$R). The performance of the HNN learning phase has

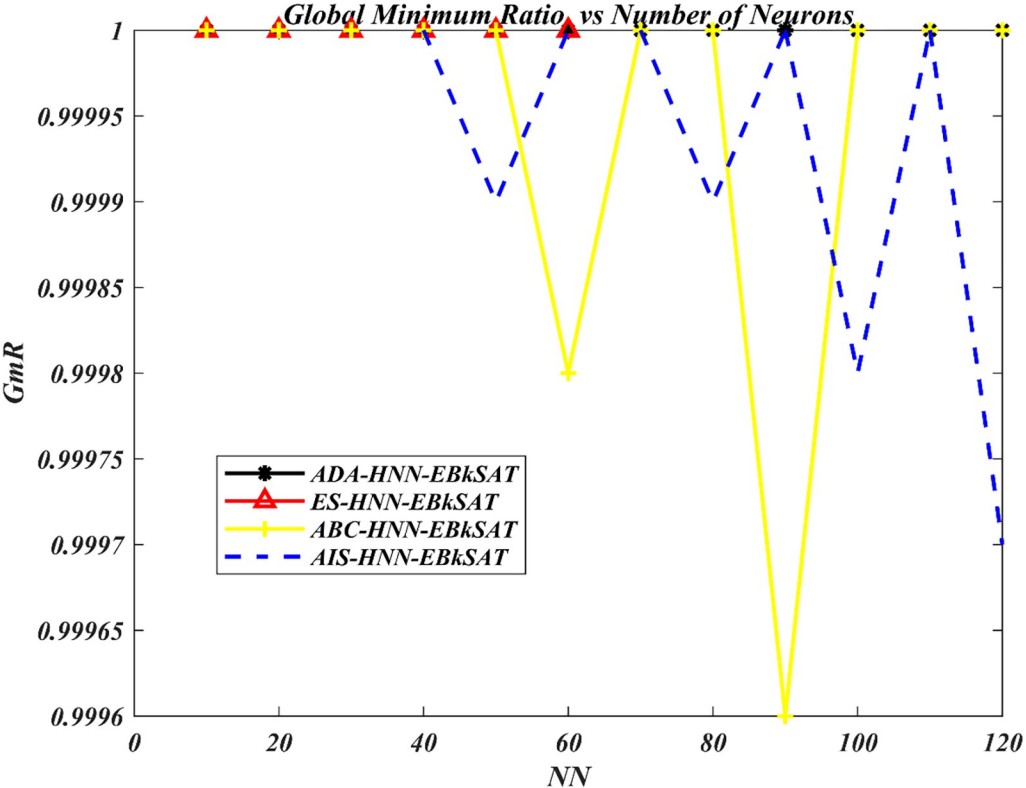

**Fig 3. G$m$R performance of various HNN-EB$k$SAT models.**

been reported 10≤*NN*≤120. The efficiency of a hybrid model can be assessed by examining their GmR for different levels of complexity in the network's neurons. Based on the results presented in Fig 3, it is evident that the ADA-HNN-EB*k*SAT and ES-HNN-EB*k*SAT models achieved more accurate neural states compared to the ABC-HNN-EB*k*SAT and AIS-HNN-EB*k*SAT models. Notably, the ES-HNN-EB*k*SAT model employed an exhaustive trial-and-error search process to ensure compliance with the clauses, which only accommodated the maximum *NN*≤60 value due to the nature of the exhaustive search technique. This exhaustive process increased the computational pressure to reach the exact neural configuration. The relationship between G*m*R and the energy state at the end of a computation cycle has been explained in [22].

Hypothetically, if a G*m*R for a hybrid network is quite near to 1, all the solutions of that system can almost achieve global minimum energy (i.e., 100 per cent satisfied clauses). It was observed that ABC-HNN-EB*k*SAT and AIS-HNN-EB*k*SAT were associated with some drawbacks, which included the tendency of being trapped at a sub-optimal output weight and a slow rate of convergence. Fig 3 describes that ABC-HNN-EB*k*SAT, neurons state was trapped at NN = 60 and 90, this is due to neurons neuron oscillations in the network searching process. AIS-HNN-EB*k*SAT recorded a high number of neurons in suboptimal solution (wrong pattern), which are located at NN = 50,80,100 and 120 neuron states. However, higher than 98% were successfully reported. On the other hand, the proposed ADA-HNN-EB*k*SAT model recorded better efficiency in the process of neuro-searching for EB*k*SAT logic representation, which is close to the existing models. As the neurons' number increases in terms of the network's complexity, this network will become more difficult as the limitations extend indefinitely. ADA-HNN-EB*k*SAT model was able to sustain more accurate neuron states than other models. This is due to the searching capacity of ADA which reduces the complexity of the network in searching for the correct EB*k*SAT representation.

In addition, the ADA strategy creates lower diversification for the $E_{F_{EBkSAT}}$ clauses, making it challenging for early solutions to fit. Therefore, ADA utilizes optimization mechanisms such as separation, cohesion, alignment, a food factor, and an enemy factor to achieve $E_{F_{EBkSAT}} = 0$ optimal solutions. The incorporation of ADA effectively reduces the learning complexity $E_{F_{EBkSAT}}$ as the number of neurons increases during simulation. The success of ADA-HNN-EBk-SAT in reaching global solutions can be attributed to its efficacy in global and local search processes, acting as a learning algorithm. In comparison to other algorithms (ABC, AIS, and ES), ADA demonstrates inspiring local search capabilities throughout the initial and final stages of the search [63–65]. This exploration highlights the robustness and efficiency of ADA's global and local search capabilities in accelerating the HNN learning process for optimal EB*k*SAT representation.

In Figs 4 and 5, the trend of ES-HNN-EB*k*SAT shows a continuous increase in error accumulation. This can be attributed to the brute force approach used in the search for satisfiability mapping out of 10≤*NN*≤60, which struggles to handle the complexity of the escalating number of neurons. On the other hand, the proposed hybrid searching method incorporating ADA mechanisms explores improved efficiency in searching for optimal EBkSAT representation, leading to $E_{F_{EBkSAT}} \rightarrow 0$. This is achieved through the intelligent search mechanisms of ADA operators, including separation, alignment, and cohesion, which facilitate the assignment that leads to $E_{F_{EBkSAT}} \rightarrow 0$. The inclusion of multiple optimization layers enables the model to reach a satisfying assignment, ultimately leading to $E_{F_{EBkSAT}} \rightarrow 0$. Additionally, a food search and enemy avoidance layer filters out non-improving solutions throughout the learning phase of HNN. The limitations of the ES-HNN-EB*k*SAT model are evident in its accumulation of high RMSE and MAPE during the learning stage, as well as its slow convergence rate, which

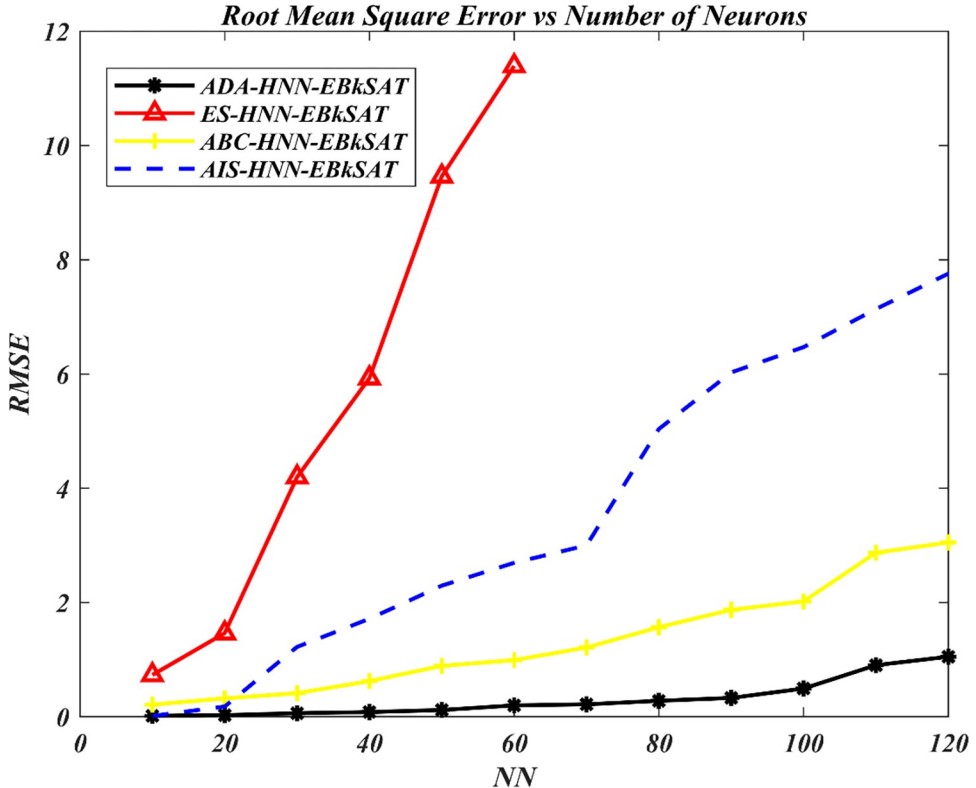

**Fig 4. RMSE performance of various HNN-EB*k*SAT models.**

requires more iterations to achieve global convergence compared to ADA-HNN-EB*k*SAT, ABC-HNN-EB*k*SAT, and AIS HNN-EB*k*SAT models.

The error analysis presented in Figs 4 and 5 reveals that ADA-HNN-EBkSAT achieves a lower RMSE and MAPE, approximately 20% lower than ABC-HNN-EB*k*SAT and 28% lower than AIS HNN-EB*k*SAT. This demonstrates the capability of ADA in reducing the model's sensitivity to errors by minimizing iterations. ADA-HNN-EB*k*SAT incorporates multiple optimization strategies to strike a balance between the exploitation and exploration aspects of the search process.

To explore the search capacity of HNN, an alignment strategy is utilized, while a cohesion strategy is employed to exploit the HNN search space, resulting in $E_{F_{EBkSAT}} \to 0$. Additionally, to facilitate the transition between exploitation and exploration, enhancing the HNN search capacity, the radii of the neighbourhood are proportionally increased with the number of iterations. Adapting swarming weights throughout the optimization process represents another approach for balancing exploitation and exploration. The best and worst EB*k*SAT clauses obtained thus far serve as the sources of food and the enemy, respectively. This mechanism promotes convergence toward a promising area $E_{F_{EBkSAT}} \to 0$ and divergence away from non-promising regions within the solution space. These ADA features compel the HNN-EB*k*SAT model to reduce the number of iterations during the learning phase, ensuring minimal error accumulation upon the completion of a computational cycle.

This systematic optimization partition in the ADA improved the local and global search process of HNN for optimal EB*k*SAT representation. This partition of a solution space allowed the model to search for the proper assignment effectively in the entire solution space, which

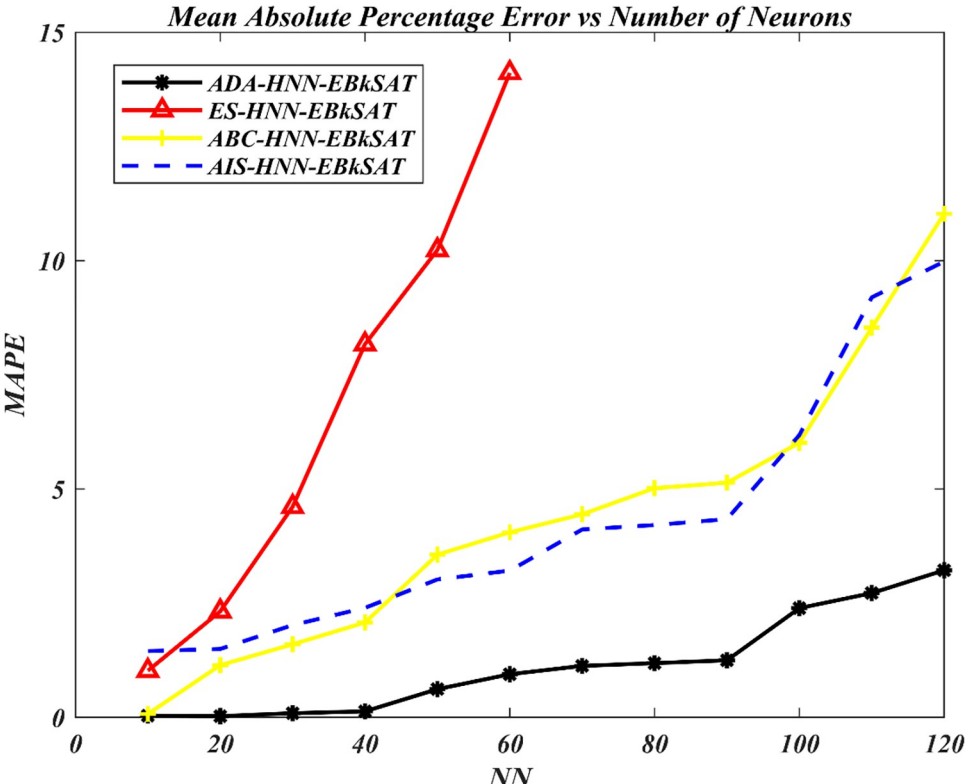

**Fig 5.** *MAPE performance of various* HNN-EB*k*SAT **models.**

led to $E_{F_{EBkSAT}} \to 0$ the lowest possible error. Specifically, the search area for ADA-HNN-EB*k*-SAT is presented as 5 spaces. ABC-HNN-EB*k*SAT and AIS-HNN-EB*k*SAT, on the contrary, with only two partition spaces and ES-HNN-EB*k*SAT with none for the overall solution space, which leads to generating a nonfit solution throughout the model's early stages of the search process, utilizing a trial-and-error method, which necessitates further iterations for obtaining a global solution. Regarding the evaluation of RMSE and MAPE, ADA can be regarded as an appropriate approach in the HNN to carry out EB*k*SAT logic representation successfully.

In Fig 6, the trend of implementation time (CT) has been displayed for all models under study. The observed running time showed that this program became more complex, taking more considerable effort and time to search for a global solution. All these models under study displayed a close time range in the search, i.e., 10≤*NN*≤40. ES-HNN-EB*k*SAT reported having consumed more implementation time in the search, i.e., 20≤*NN*≤60, which made it slower than other models. ES-HNN-EB*k*SAT was observed to have consumed 862 seconds slower than that of ADA-HNN-EB*k*SAT and ABC-HNN-EB*k*SAT model recorded approximately 724 seconds slower than that of AIS-HNN- EB*k*SAT. According to Fig 6, ADA-HNN-EB*k*SAT required less time to executant compared with ABC-HNN-EB*k*SAT and AIS HNN-EB*k*SAT models. This means that when the neurons' number increases, the time accumulation will be more. In addition, ADA-HNN-EBkSAT required fewer iterations to find the desired solution, resulting in reduced execution time. On the other hand, ES-HNN-EB*k*SAT needed more iterations to find the global solution, as the interpretation collapses when optimal fitness cannot be achieved, leading to a new interpretation hunt with a different fitness value. Hence, ADA has proven to be more effective in enhancing or pursuing preferred approaches, even in scenarios with high complexity literals. This can be attributed to the increased number of layers in the

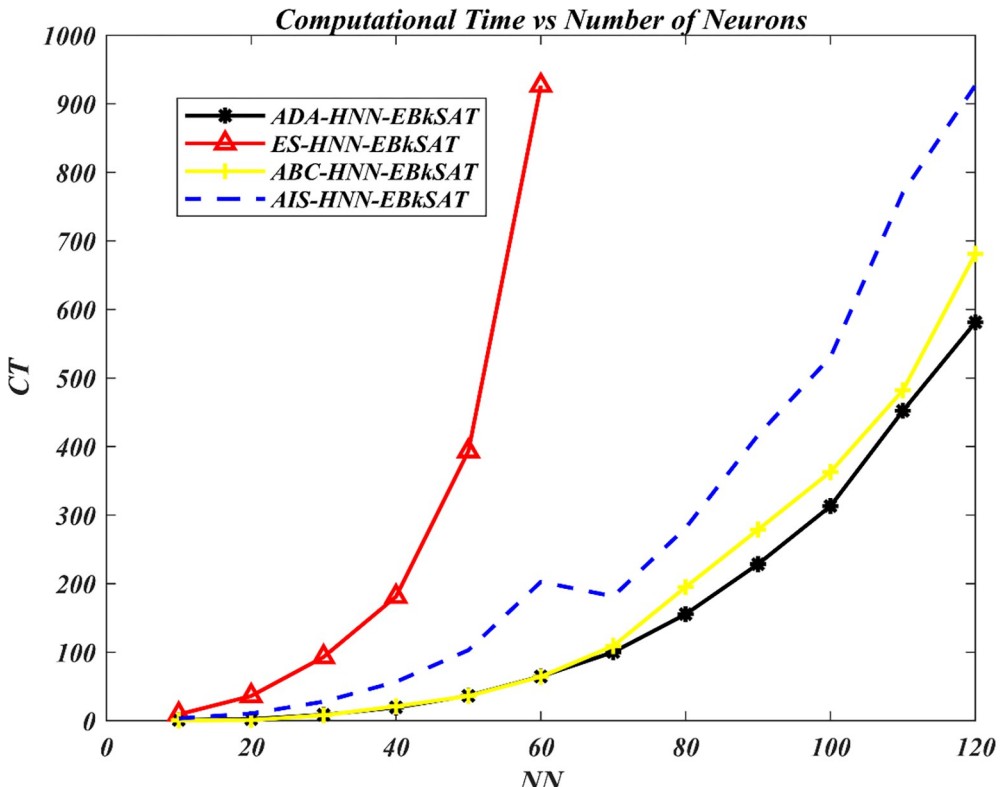

**Fig 6. Implementation time performance of various HNN-EBkSAT models.**

ADA searching process, allowing for the resolution of greater eligibility in a shorter time frame.

These ADA features have enabled HNN-EBkSAT to complete the learning phase more quickly compared to current models in the literature. It has been verified that ADA-HNN-EBkSAT achieved this learning process with a slightly shorter timeframe than other existing models. However, all HNN models demonstrated competence in optimizing EBkSAT and its variants, successfully computing a global solution within a feasible CPU time.

Table 4 presents the overall testing error and accuracy of different models for a classification problem. The results demonstrate that the proposed logical rule, EBkSAT, consistently provides optimal classification to the HNN during the learning phase, resulting in very low error. The training error, represented by MAPE and RMSE, indicates the deviation between predicted and actual outputs during training. Lower values of MAPE and RMSE indicate better accuracy and reduced errors in training.

In terms of MAPE, the ADA-HNN-EBkSAT model achieved the lowest training error of 7.65, followed by the ABC-HNN-EBkSAT model with a slightly higher MAPE of 10.93. The

**Table 4. Training error and accuracy for all HNN models.**

| Logical rule | MAPE | RMSE | ACCURACY |
|---|---|---|---|
| ADA-HNN-EBkSAT | 7.65 | 3.47 | 93.1 |
| ABC-HNN-EBkSAT | 10.93 | 4.93 | 89.3 |
| AIS-HNNkSAT | 12.17 | 8.16 | 87.5 |
| ES-HNNkSAT | 21.16 | 12.88 | 82.5 |

**Table 5. Comparison of MAPE mean and RMSE mean for statistical significance.**

| Comparison | MAPE Mean | RMSE Mean |
|---|---|---|
| ADA-HNN-EBkSAT and ABC-HNN-EBkSAT | p < 0.05 (statistically significant) | p < 0.05 (statistically significant) |
| ADA-HNN-EBkSAT and AIS-HNN-EBkSAT | p < 0.05 (statistically significant) | p < 0.05 (statistically significant) |
| ADA-HNN-EBkSAT and ES-HNN-EBkSAT | p < 0.05 (statistically significant) | p < 0.05 (statistically significant) |
| ABC-HNN-EBkSAT and ES-HNN-EBkSAT | p < 0.05 (statistically significant) | p < 0.05 (statistically significant) |
| AIS-HNN-EBkSAT and ES-HNN-EBkSAT | p < 0.05 (statistically significant) | p < 0.05 (statistically significant) |

AIS-HNNkSAT model had a MAPE of 12.17, and the ES-HNNkSAT model had the highest MAPE of 21.16. Similarly, in terms of RMSE, the ADA-HNN-EBkSAT model had the lowest training error of 3.47, followed by the ABC-HNN-EBkSAT model with an RMSE of 4.93. The AIS-HNNkSAT model had an RMSE of 8.16, and the ES-HNNkSAT model had the highest RMSE of 12.88. The accuracy measures indicate the models' performance in correctly classifying instances. The ADA-HNN-EBkSAT model achieved the highest accuracy of 93.1%, outperforming the other models. The ABC-HNN-EBkSAT, AIS-HNN-EBkSAT, and ES-HNN-EBkSAT models achieved accuracies of 89.3%, 87.5%, and 82.5% respectively. These results highlight the superior accuracy of the ADA-HNN-EBkSAT model for the given classification problem.

Statistical analysis was conducted to compare the performance of the models based on MAPE, RMSE, and accuracy. The ANOVA tests revealed significant differences in the mean MAPE, RMSE, and accuracy values among the models. Tukey's HSD test was then performed to identify specific pairwise differences between the models. The results of Tukey's HSD test in Table 5 indicate the following significant differences:

In the Table 5, the first column represents the comparison between different models or techniques, and the second column indicates the statistical significance with "p < 0.05" denoting that the difference observed between the compared models is statistically significant.

The results of this statistical analysis provide valuable insights into the effectiveness of the models for classification tasks in comparison. Researchers and practitioners can utilize this information to select the most suitable model based on their specific requirements. Additionally, conducting postdoc tests can offer a more nuanced analysis and further contribute to our understanding of the differences among the models. This analysis serves as a valuable reference for those working on classification problems and contributes to ongoing research in the field. These findings indicate significant differences in the mean MAPE and RMSE values between the ADA-HNN-EBkSAT model and the other models. The ADA-HNN-EBkSAT model consistently outperforms the ABC-HNN-EBkSAT, AIS-HNN-EBkSAT, and ES-HNN-EBkSAT models in terms of accuracy and error metrics.

## Conclusion

Based on the simulation results, the ADA-HNN-EBkSAT model exhibited superior efficiency and robustness compared to the ABC-HNN-EBkSAT, AIS-HNN-EBkSAT, and ES-HNN-EBkSAT models in accelerating the learning phase of the HNN for the EBkSAT logic program, while achieving lower learning error. The proposed model also demonstrated good agreement with the ABC-HNN-EBkSAT and AIS-HNN-EBkSAT models in terms of G$m$R, RMSE, MAPE, CT, and accuracy metrics.

Notably, the ADA-HNN-EBkSAT model achieved a G$m$R of 1, even when handling complex network structures, indicating its efficacy in approaching the global minimum. This highlights the effectiveness of the ADA algorithm as a powerful heuristic for enhancing the training phase of the HNN in the context of the EBkSAT logic program.

However, it is important to acknowledge certain practical and theoretical limitations in this study. Firstly, the evaluation was conducted based on simulated data sets, and the performance of the proposed model should be further validated using real-world datasets to ensure its generalizability. Additionally, the comparison was limited to specific existing models, and incorporating a broader range of benchmark models would provide a more comprehensive evaluation. Furthermore, although the ADA-HNN-EB$k$SAT model showed promising results in terms of accuracy and computational time, there may be specific problem instances or scenarios where alternative metaheuristic methods could yield better performance. Exploring and incorporating other metaheuristic approaches in future research would enable a more comprehensive understanding of the potential acceleration of the HNN computational phase.

Finally, the ADA-HNN-EB$k$SAT model demonstrated its superiority in accelerating the learning phase of the HNN for the EB$k$SAT logic program, outperforming existing models in terms of efficiency and accuracy. While acknowledging the practical and theoretical limitations of the study, the findings suggest the potential practical applicability of the proposed model in various domains where complex optimization problems are encountered.

## Acknowledgments

The authors are thankful to the Deanship of Scientific Research at Najran University and the Registrar of Universiti Tun Hussein Onn Malaysia.

## Author Contributions

**Conceptualization:** Hamza Abubakar, Shehab Abdulhabib Saeed Alzaeemi, Kim Gaik Tay.

**Funding acquisition:** Shehab Abdulhabib Saeed Alzaeemi.

**Methodology:** Ghassan Ahmed Ali.

**Software:** Hamza Abubakar.

**Writing – original draft:** Abdulkarem H. M. Almawgani.

**Writing – review & editing:** Hamza Abubakar, Adel Sulaiman, Kim Gaik Tay.

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
