## [Decision Letter · Decision Letter 0]

20 Mar 2023

PONE-D-22-26132ARTIFICIAL DRAGONFLY ALGORITHM IN THE HOPFIELD NEURAL NETWORK FOR  BOOLEAN k SATISFIABILITY REPRESENTATIONPLOS ONE

Dear Dr. ABUBAKAR,

Thank you for submitting your manuscript to PLOS ONE. After careful consideration, we feel that it has merit but does not fully meet PLOS ONE’s publication criteria as it currently stands. Therefore, we invite you to submit a revised version of the manuscript that addresses the points raised during the review process.

ACADEMIC EDITOR: Dear authors, please carefully revise proposed manuscript according to all reviewers' comments and my comments. Additionally, according to PLOSONE journal policy, please make sure that your source code is available. I suggest you to use the data archiving tool Zenodo and generate doi which will be included in the revised version of the manuscript. 

We look forward to receiving your revised manuscript.

Kind regards,

Nebojsa Bacanin

Academic Editor

PLOS ONE

“The authors are thankful to the Deanship of Scientific Research at Najran University for funding this work under the Research Collaboration Funding program grant coder NU/RC/SERC/11/5 and they are thankful to the registrar Univeristi Tun Hussein Onn Malaysia.”

3. Thank you for stating the following in the Acknowledgments/ Funding Section of your manuscript:

“The authors are thankful to the Deanship of Scientific Research at Najran University for funding this work under the Research Collaboration Funding program grant coder NU/RC/SERC/11/5 and they are thankful to the registrar Univeristi Tun Hussein Onn Malaysia.”

 “The authors are thankful to the Deanship of Scientific Research at Najran University for funding this work under the Research Collaboration Funding program grant coder NU/RC/SERC/11/5 and they are thankful to the registrar Univeristi Tun Hussein Onn Malaysia.”

“The authors are thankful to the Deanship of Scientific Research at Najran University for funding this work under the Research Collaboration Funding program grant coder NU/RC/SERC/11/5 and they are thankful to the registrar Univeristi Tun Hussein Onn Malaysia”

Additional Editor Comments:

Dear Authors,

please revise proposed manuscript thoroughly according to all reviewers' comments.

Additionally, please do the following:

- In introduction section you must clearly state what is beyond state-of-the-art in proposed study.

- Statistical tests must be conducted to validate that the improvements of proposed approach are statistically significant in terms of employed performance metrics.

- Figures qualities must be improved. For examples, figs 1 and 2 look like they were taken from other source as it is. If this is the case, please generated your own images.

- I would suggest including more methods in comparative analysis.

- In the conclusion section you should clearly state what are practical and theoretical implications, as well as limitations of proposed study.

- References must be updated with more recent and relevant ones.

All the best,

Nebojsa Bacanin

Reviewers' comments:

Reviewer's Responses to Questions

**Comments to the Author**

1. Is the manuscript technically sound, and do the data support the conclusions?

Reviewer #1: Yes

Reviewer #2: Partly

2. Has the statistical analysis been performed appropriately and rigorously? 

Reviewer #1: No

Reviewer #2: Yes

3. Have the authors made all data underlying the findings in their manuscript fully available?

Reviewer #1: Yes

Reviewer #2: No

4. Is the manuscript presented in an intelligible fashion and written in standard English?

Reviewer #1: Yes

Reviewer #2: Yes

5. Review Comments to the Author

Reviewer #1: 1. Is there any particular reason why the entire text is highlighted in yellow? It made reading very hard.

2. Avoid using acronyms in the abstract.

3. Introduction is too long. Separate it into Introduction and Literature review, two distinctive sections.

4. Paragraphs are too long as well, consider breaking them in smaller units.

5. State the research question clearly in the Introduction.

6. List the main contributions clearly in the Introduction (they are present, but should be emphasized more, for example like a list).

7. The literature review could be expanded, consider adding the following references that deal with dragonfly algorithm, and also ANN optimization by metaheuristics:

https://www.sciencedirect.com/science/article/abs/pii/S2210537922000506

https://www.sciencedirect.com/science/article/abs/pii/S0003682X21002152

https://link.springer.com/chapter/10.1007/978-981-15-8530-2_63

https://link.springer.com/article/10.1007/s00521-022-06946-7

https://www.mdpi.com/1996-1073/16/3/1434

8. Make sure that each parameter in every equation has been explained in the text.

9. Overall quality of figures should be improved - especially flowchart (fig. 2).

10. Discuss more why you have selected dragonfly - as there are numerous other metaheuristics algorithms.

11. Discuss the limitations of the proposed method in more details.

12. Elaborate more on the simulation setup - also give the results in tabular form (not only by graphs and charts).

Reviewer #2: In this paper, a hybrid computational method combining the artificial dragonfly algorithm (ADA) with the Hopfield neural network techniques for optimal Exact k Satisfiability representation is presented.

Several issues in this paper need to be addressed before publication:

The paper requires some reorganization. In the Introduction section, it is important to provide a brief overview of the topic addressed in the paper, along with a concise description of the main contributions, which are currently listed but need to be explained in greater detail. Additionally, a new section entitled "Related Work" should be added to the paper, which will include a description of previously used methodologies and systems that are currently part of the Introduction section.

Furthermore, it is essential to enhance the quality of Figure 2. Additionally, Figure 6 contains typos that need to be corrected.

Regarding the evaluation, it is important to note that simulated datasets were used to establish the Exact kSAT logical clauses. However, further explanation is required to fully understand this process. Therefore, additional effort is needed to test and prove the set hypotheses on the available datasets.

6. PLOS authors have the option to publish the peer review history of their article (what does this mean?). If published, this will include your full peer review and any attached files.

Reviewer #1: No

Reviewer #2: No

---

## [Author Response · Author response to Decision Letter 0]

27 Apr 2023

----Reviewer #1:---

Reviewer(s)' Comments to Author

1. Is there any particular reason why the entire text is highlighted in yellow? It made reading very hard.

Response to Reviewer(s)' Comments

It is the revised version. We sorry about that. Thank you very much

Reviewer(s)' Comments to Author

2. Avoid using acronyms in the abstract.

Response to Reviewer(s)' Comments

Thank you very much for this comment. All acronyms has been removed accordingly. 

Reviewer(s)' Comments to Author

3. Introduction is too long. Separate it into Introduction and Literature review, two distinctive sections.

Response to Reviewer(s)' Comments

This section has been separated accordingly.

Reviewer(s)' Comments to Author

4. Paragraphs are too long as well, consider breaking them in smaller units.

Response to Reviewer(s)' Comments

This section has been revised accordingly. Thank you very much for comments

Reviewer(s)' Comments to Author

5. State the research question clearly in the Introduction.

Response to Reviewer(s)' Comments

Thank you very much for this comments. We have addeded research question accordongly

Reviewer(s)' Comments to Author

6. List the main contributions clearly in the Introduction (they are present, but should be emphasized more, for example like a list).

Response to Reviewer(s)' Comments

The main contributions is clearly stated. Thank you very much for this comments

Reviewer(s)' Comments to Author

7. The literature review could be expanded, consider adding the following references that deal with dragonfly algorithm, and also ANN optimization by metaheuristics:

https://www.sciencedirect.com/science/article/abs/pii/S2210537922000506

https://www.sciencedirect.com/science/article/abs/pii/S0003682X21002152

https://link.springer.com/chapter/10.1007/978-981-15-8530-2_63

https://link.springer.com/article/10.1007/s00521-022-06946-7

https://www.mdpi.com/1996-1073/16/3/1434

Response to Reviewer(s)' Comments

Thank you very much. We have cited theses articles accordingly

Reviewer(s)' Comments to Author

8. Make sure that each parameter in every equation has been explained in the text.

Response to Reviewer(s)' Comments

All parameters have been explained accordingly.

Reviewer(s)' Comments to Author

9. Overall quality of figures should be improved - especially flowchart (fig. 2).

Response to Reviewer(s)' Comments

Thank you very much for this comment. We have replotted all figures with high resolutions.

Reviewer(s)' Comments to Author

10. Discuss more why you have selected dragonfly - as there are numerous other metaheuristics algorithms.

Response to Reviewer(s)' Comments

Thank you very much. There is no specifi reason for this selction only to explored the performance of dragonfly

Reviewer(s)' Comments to Author

11. Discuss the limitations of the proposed method in more details.

Response to Reviewer(s)' Comments

We have added discussion accordingly.

Reviewer(s)' Comments to Author

12. Elaborate more on the simulation setup - also give the results in tabular form (not only by graphs and charts).

Response to Reviewer(s)' Comments

Thank you for the constructive comments, we added table to show the best result as requste by the reviewer.

----Reviewer #2: ----

Reviewer(s)' Comments to Author

In this paper, a hybrid computational method combining the artificial dragonfly algorithm (ADA) with the Hopfield neural network techniques for optimal Exact k Satisfiability representation is presented.

Reviewer(s)' Comments to Author

Several issues in this paper need to be addressed before publication:

The paper requires some reorganization. In the Introduction section, it is important to provide a brief overview of the topic addressed in the paper, along with a concise description of the main contributions, which are currently listed but need to be explained in greater detail. Additionally, a new section entitled "Related Work" should be added to the paper, which will include a description of previously used methodologies and systems that are currently part of the Introduction section.

Response to Reviewer(s)' Comments

Thank you for the constructive comments, we added the related work at section of literature review as requste by the reviewer.

Reviewer(s)' Comments to Author

Furthermore, it is essential to enhance the quality of Figure 2. Additionally, Figure 6 contains typos that need to be corrected.

Response to Reviewer(s)' Comments

Thank you for the constructive comments, we refigure all figures.

Reviewer(s)' Comments to Author

Regarding the evaluation, it is important to note that simulated datasets were used to establish the Exact kSAT logical clauses. However, further explanation is required to fully understand this process. Therefore, additional effort is needed to test and prove the set hypotheses on the available datasets.

Response to Reviewer(s)' Comments

in this work we use simulated data, Simulated data is a randomly generated data by our program during the simulation. The generated data considers the binary values with the structure based on the number of clauses defined by the researchers. Thus, the simulated data is commonly used to authenticate the performance our proposed networks in discriminating the solutions, whether it is an optimal or suboptimal solution. Therefore, the database is massive, and it covers a more diverse search space. The concept of simulated data in SAT logic with ANN has been initially coined in the work of Saratha, S. (2010) in HNN development and Abdulhabib et al. (2021).

---

## [Decision Letter · Decision Letter 1]

16 May 2023

PONE-D-22-26132R1ARTIFICIAL DRAGONFLY ALGORITHM IN THE HOPFIELD NEURAL NETWORK FOR  BOOLEAN k SATISFIABILITY REPRESENTATIONPLOS ONE

Dear Dr. ABUBAKAR,

Thank you for submitting your manuscript to PLOS ONE. After careful consideration, we feel that it has merit but does not fully meet PLOS ONE’s publication criteria as it currently stands. Therefore, we invite you to submit a revised version of the manuscript that addresses the points raised during the review process.

ACADEMIC EDITOR: 

Dear Authors,

thank you for improving your manuscript, however I still have some comments.

- Figure quality should be improved, e.g. x-axis titles are too large and they should be resized.

- There are many inconsistencies in titles numbering, e.g. "1. Materials and Methods 2.1 Exact k Satisfiability of a Boolean Formula".

- Please make sure that all notations in expressions are defined.

- Please conduct appropriate statistical analysis. 

- Conclusion should be extended to include practical and theoretical limitations of proposed study.

- Please make sure that you ad here with all PLOS ONE policies. 

Thanks.

Warmest,

AE

We look forward to receiving your revised manuscript.

Kind regards,

Nebojsa Bacanin

Academic Editor

PLOS ONE

Journal Requirements:

Additional Editor Comments:

Dear Authors,

thank you for improving your manuscript, however I still have some comments.

- Figure quality should be improved, e.g. x-axis titles are too large and they should be resized.

- There are many inconsistencies in titles numbering, e.g. "1. Materials and Methods 2.1 Exact k Satisfiability of a Boolean Formula".

- Please make sure that all notations in expressions are defined.

- Please conduct appropriate statistical analysis.

- Conclusion should be extended to include practical and theoretical limitations of proposed study.

- Please make sure that you ad here with all PLOS ONE policies.

Thanks.

Warmest,

AE

Reviewers' comments:

Reviewer's Responses to Questions

**Comments to the Author**

1. If the authors have adequately addressed your comments raised in a previous round of review and you feel that this manuscript is now acceptable for publication, you may indicate that here to bypass the “Comments to the Author” section, enter your conflict of interest statement in the “Confidential to Editor” section, and submit your "Accept" recommendation.

Reviewer #1: All comments have been addressed

Reviewer #2: All comments have been addressed

2. Is the manuscript technically sound, and do the data support the conclusions?

Reviewer #1: (No Response)

Reviewer #2: Yes

3. Has the statistical analysis been performed appropriately and rigorously? 

Reviewer #1: (No Response)

Reviewer #2: Yes

4. Have the authors made all data underlying the findings in their manuscript fully available?

Reviewer #1: (No Response)

Reviewer #2: Yes

5. Is the manuscript presented in an intelligible fashion and written in standard English?

Reviewer #1: (No Response)

Reviewer #2: Yes

6. Review Comments to the Author

Reviewer #1: (No Response)

Reviewer #2: All of the requirements were met by the authors. The new version of the manuscript meets the requirements for publication in this type of journal.

7. PLOS authors have the option to publish the peer review history of their article (what does this mean?). If published, this will include your full peer review and any attached files.

Reviewer #1: No

Reviewer #2: No

---

## [Author Response · Author response to Decision Letter 1]

20 May 2023

Response to Reviewer(s)' Comments

Thank you for the constructive comments, we have update our paper better as requested by reviewers.

---- ACADEMIC EDITOR :---

Academic (s)' Comments to Author

- Figure quality should be improved, e.g. x-axis titles are too large and they should be resized.

Response To Academic Editor(S)' Comments

The quality of figures have been improve accordingly. Thank you very for this comment.

Academic (s)' Comments to Author

- There are many inconsistencies in titles numbering, e.g. "1. Materials and Methods 2.1 Exact k Satisfiability of a Boolean Formula".

Response To Academic Editor(S)' Comments

It is sincerely regarrted. All inconsistenies have been checked and corrected accoringly

Academic (s)' Comments to Author

- Please make sure that all notations in expressions are defined.

Response To Academic Editor(S)' Comments

All notations in expressions have been defined accordingly.

Academic (s)' Comments to Author

- Please conduct appropriate statistical analysis.

Response To Academic Editor(S)' Comments

Thank you very much for this comment. This studied used RMSE and MAPE and Accuacy as as appropriate statistical measured. However, we have added Tukey's HSD test for further analysis.

Academic (s)' Comments to Author

- Conclusion should be extended to include practical and theoretical limitations of proposed study.

Response To Academic Editor(S)' Comments

Thank you very much. Some of the practical and theoretical limitations of proposed study have been highlighlited accordingly.

Academic (s)' Comments to Author

- Please make sure that you ad here with all PLOS ONE policies

Response To Academic Editor(S)' Comments

It is the revised version. We are sorry about that. Thank you very much

---

## [Editor Report · Decision Letter 2]

25 May 2023

ARTIFICIAL DRAGONFLY ALGORITHM IN THE HOPFIELD NEURAL NETWORK FOR  BOOLEAN k SATISFIABILITY REPRESENTATION

PONE-D-22-26132R2

Dear Dr. ABUBAKAR,

We’re pleased to inform you that your manuscript has been judged scientifically suitable for publication and will be formally accepted for publication once it meets all outstanding technical requirements.

Kind regards,

Nebojsa Bacanin

Academic Editor

PLOS ONE

Additional Editor Comments (optional):

Dear Authors,

thank you for revising your manuscript.

Only one minor note: I think that the Tukey's HSD test analysis results should be in tabular form.

All the best,

AE
---

## [Editor Report · Acceptance letter]

13 Jun 2023

PONE-D-22-26132R2 

ARTIFICIAL DRAGONFLY ALGORITHM IN THE HOPFIELD NEURAL NETWORK FOR OPTIMAL EXACT BOOLEAN *k* SATISFIABILITY REPRESENTATION 

Dear Dr. Abubakar:

I'm pleased to inform you that your manuscript has been deemed suitable for publication in PLOS ONE. Congratulations! Your manuscript is now with our production department. 

Kind regards, 

on behalf of

Dr. Nebojsa Bacanin 

Academic Editor

PLOS ONE